# Beyond Instance Consistency:
# Investigating View Diversity in Self-supervised Learning

**Huaiyuan Qin**                                            *qinhy@i2r.a-star.edu.sg*
*Institute for Infocomm Research ($I^2R$), A*STAR, Singapore*

**Muli Yang**                                               *yangml@i2r.a-star.edu.sg*
*Institute for Infocomm Research ($I^2R$), A*STAR, Singapore*

**Siyuan Hu**                                               *siyuanhu@nus.edu.sg*
*National University of Singapore, Singapore*

**Peng Hu**                                                 *penghu.ml@gmail.com*
*Sichuan University, China*

**Yu Zhang**                                                *zhang_yu@seu.edu.cn*
*Southeast University, China*

**Chen Gong**                                               *chen.gong@sjtu.edu.cn*
*Shanghai Jiaotong University, China*

**Hongyuan Zhu**[*]                                         *zhuh@i2r.a-star.edu.sg*
*Institute for Infocomm Research ($I^2R$), A*STAR, Singapore*

**Reviewed on OpenReview:** *https://openreview.net/forum?id=urWCU3YMA0*

## Abstract

Self-supervised learning (SSL) conventionally relies on the instance consistency paradigm, assuming that different views of the same image can be treated as positive pairs. However, this assumption breaks down for non-iconic data, where different views may contain distinct objects or semantic information. In this paper, we investigate the effectiveness of SSL when instance consistency is not guaranteed. Through extensive ablation studies, we demonstrate that SSL can still learn meaningful representations even when positive pairs lack strict instance consistency. Furthermore, our analysis further reveals that increasing view diversity, by enforcing zero overlapping or using smaller crop scales, can enhance downstream performance on classification and dense prediction tasks. However, excessive diversity is found to reduce effectiveness, suggesting an optimal range for view diversity. To quantify this, we adopt the Earth Mover's Distance (EMD) as an estimator to measure mutual information between views, finding that moderate EMD values correlate with improved SSL learning, providing insights for future SSL framework design. We validate our findings across a range of settings, highlighting their robustness and applicability on diverse data sources.

## 1 Introduction

Humans can effortlessly recognize objects across different viewpoints and contexts. A cat lounging on a couch remains a cat, whether seen from the side or above. This *identity-invariant* consistency has inspired the design of self-supervised learning (SSL) methods in computer vision, which leverage cross-view consistency as a supervision signal (Jing & Tian, 2020; He et al., 2020; Chen et al., 2020b; 2021; Grill et al., 2020; Caron

---

[*]Corresponding author.

(a) Random Crops on *Iconic* Data  (b) Random Crops on *Non-iconic* Data

Figure 1: **Visualization of random crops on *iconic* data (*e.g.* ImageNet (Deng et al., 2009)) and *non-iconic* data (*e.g.* COCO (Lin et al., 2014)).** For *iconic* data, different views of the same image maintain instance consistency. However, for *non-iconic* data, different views may capture entirely different object instances, leading to the breakdown of such consistency.

et al., 2020; 2021; Oquab et al., 2023; Siméoni et al., 2025). SSL has emerged as an effective approach for learning visual representations from unlabeled data by aligning different views of the same image, relying on the instance consistency paradigm (Wu et al., 2018), which considers each image as a separate class. In this paradigm, different augmentations of an image, such as cropping, rotation, or color jittering, are treated as positive pairs, while augmentations from other images serve as negatives (He et al., 2020; Chen et al., 2020a). The goal is to learn representations that capture essential semantic information from common instance while discarding irrelevant variations.

This instance consistency paradigm works remarkably well for *iconic* datasets, where images typically feature a single, dominant object, ensuring that different views naturally share the same semantic content of the common instance. However, training on such iconic datasets like ImageNet (Deng et al., 2009) poses challenges in scalability, due to the requirement of intensive data collection and cleaning. In contrast, *non-iconic* datasets, such as COCO (Lin et al., 2014) and OpenImages (Kuznetsova et al., 2020), are easier to collect but introduce a fundamental challenge: these non-iconic datasets often feature complex scenes with multiple objects and diverse backgrounds (Van Gansbeke et al., 2021; Selvaraju et al., 2021; Zhu et al., 2023; Chuang et al., 2022; Stegmüller et al., 2023; Mishra et al., 2022), leading to the facts that two augmented views from the same image may not guarantee to contain the same object or share consistent semantic information, as illustrated in Figure 1. Surprisingly, despite the breakdown of instance consistency on non-iconic data, SSL methods can still achieve competitive performance, as reported in Van Gansbeke et al. (2021), Mishra et al. (2022) and Zhu et al. (2023). This challenges the conventional assumption that positive pairs must always share instance semantics, raising a critical research question: **Is instance consistency a strict requirement for self-supervised learning?**

To address this, we investigate in the following two key aspects: (1) **How does SSL perform under different levels of instance consistency?** We systematically evaluate whether SSL can still function effectively when positive pairs contain minimal shared instance semantics. Our configurations range from overlapping views with shared instance and background patterns, to entirely non-overlapping views with limited shared foreground content. We also explore configurations where only background information is shared or one view contains foreground while the other contains only background. Surprisingly, our results reveal that strict instance consistency is less essential in SSL than previously assumed. To further explore this observation, we study: (2) **How much diversity between positive pairs is beneficial, and when does it become detrimental?** We observe that increasing the diversity between positive pairs, such as enforcing zero overlapping or using smaller crop scales, could encourage the model to discover more fine-grained visual consistencies, particularly benefiting classification and dense prediction tasks. However, excessively increasing the diversity in between can hinder the effectiveness, leading to a performance drop in downstream evaluations. This suggests that an optimal range of the shared information exists, where the balance between the consistency and diversity in positive pairs plays a significant role for effective SSL.

To quantify this balance, we adopt Earth Mover's Distance (EMD) as a metric to measure the view diversity. Our analysis reveals that moderate EMD values correlate with improved SSL performance, providing a useful estimator for guiding positive pair selection in future SSL framework design. Finally, we validate our findings

above across a range of settings, including multiple SSL methods, various training datasets, and a broad range of downstream evaluation tasks, demonstrating the practical applicability of our insights for effective SSL across diverse application scenarios.

In short, our main contributions are as follows:

1. **Revisiting the Necessity of Instance Consistency**: We empirically demonstrate that strict instance consistency is not essential for effective SSL, as models can leverage broader contextual cues even when positive pairs contain minimal shared instance semantics. Meanwhile, we show that increasing diversity between positive pairs can encourage the discovery of fine-grained visual consistencies, enhancing SSL's effectiveness. However, excessive diversity can hinder learning, suggesting the existence of an optimal range for view diversity.

2. **Earth Mover's Distance as an Estimator for Optimal View Diversity**: We adopt Earth Mover's Distance (EMD) as an estimator to quantify mutual information between positive pairs, finding it to be a predictive measure of view diversity for effective SSL.

3. **Validation Across Diverse Methods, Datasets and Tasks**: We validate our findings across diverse settings, while also demonstrating the robustness and generalizability of our proposed EMD-based diversity estimator on various data sources.

## 2 Related Work

**Self-supervised Learning.** Self-supervised learning (SSL) has emerged as a powerful technique for learning rich visual representations from unlabeled data (Jing & Tian, 2020), which have demonstrated impressive performance improvements across various downstream tasks (Caron et al., 2020; Grill et al., 2020; Oquab et al., 2023; Darcet et al., 2023; Henaff, 2020; Luo et al., 2023; Li et al., 2020; Ma et al., 2022; Bardes et al., 2021; Chen et al., 2020a; 2023a;b; Siméoni et al., 2025).

Early SSL efforts primarily focused on pretext tasks that leverage image-level or spatial-level structures to create supervisory signals. These include jigsaw puzzle solving (Noroozi & Favaro, 2016), where a network learns to rearrange shuffled patches into the correct spatial configuration; RotNet (Gidaris et al., 2018), which trains models to predict the rotation angle applied to an image; and video-based SSL methods (Jayaraman & Grauman, 2016), which exploit temporal coherence to learn feature representations from adjacent video frames. While foundational, these methods are often limited in scalability and generalization.

Recent SSL methods can be broadly categorized into three main groups: (1) Contrastive learning based SSLs (Chen et al., 2020a;b; 2021), are designed to optimize representations by maximizing the similarity between augmented views of the same image while minimizing similarity with other images. These methods typically rely on the assumption that each image represents a single object-centric entity, making them well-suited for iconic datasets like ImageNet (Deng et al., 2009). (2) Self-distillation based SSLs (Grill et al., 2020; Caron et al., 2021; Oquab et al., 2023; Caron et al., 2020; Siméoni et al., 2025), remove the need for explicit negative pairs by focusing on consistency between teacher and student networks. These approaches also need to learn by aligning representations across augmented views, making them effective in object-centric contexts but challenging to extend to complex, multi-object scenes. (3) Reconstruction-based SSLs (He et al., 2022; Xie et al., 2022b), aim to reconstruct masked parts of the image, leveraging spatial context to learn representations. These techniques inherently focus on local structures, making them suitable for tasks requiring spatial feature preservation. Both contrastive-based and distillation-based SSLs primarily rely on the instance consistency (Wu et al., 2018), where each image is treated as a separate class.

**SSL on Non-iconic Data.** Recent work has explored extending SSL to non-iconic datasets (Lin et al., 2014; Kuznetsova et al., 2020), which pose unique challenges due to complex scenes with multiple objects. Approaches like Zhao et al. (2021), Liu et al. (2020), Stegmüller et al. (2023), Chen et al. (2023b) and Wang et al. (2021) attempt to align dense features across views within a single image. However, these methods often require on manual feature- or image-level matching, which is challenging in complex scenes. Other techniques handle non-iconic data by pre-processing images into object-centric patches, using supervised (Mishra et al.,

2022; Selvaraju et al., 2021) or unsupervised (Zhu et al., 2023; Peng et al., 2022) object discovery methods, allowing traditional SSL frameworks to operate effectively on these pseudo-object-centric images. Alternative methods have introduced new loss functions designed to handle noisy or inconsistent semantics in varied views (Chuang et al., 2022). Meanwhile, studies on SSL with natural images (Goyal et al., 2021; 2022) suggest that random cropping remains broadly effective, with Van Gansbeke et al. (2021) offering empirical evidence for its applicability. In parallel, Purushwalkam & Gupta (2020) conduct an early investigation into SSL invariances and dataset biases, finding that models trained on non-iconic data perform worse than those trained on iconic data. However, a deeper analysis of view consistency and diversity remains under-explored, which is essential to fully leverage SSL's potential.

**Data Augmentation in SSL.** Data augmentation plays a fundamental role in SSL, providing the supervision signal to learn meaningful representations by capturing invariance across augmentations. Experiments in SimCLR (Chen et al., 2020a) have established a detailed ablation studies on augmentation strategies, concluding that applying diverse transformations to positive pairs improves representation learning. These findings have since become the standard practice in modern SSL methods (He et al., 2020; Caron et al., 2021; 2020; Grill et al., 2020), which typically employ a loss function that pushes together representations of augmented views. More recently, Morningstar et al. (2024) propose a unified SSL framework, showing that augmentation diversity plays a critical role in the success of recent SSL methods. Our work investigates how augmentation-induced view diversity affects SSL when positive pairs contain minimal shared semantics.

**Earth Mover's Distance.** Earth Mover's Distance (EMD) is widely used in computer vision as a metric to quantify structural similarity between distributions. Initially applied in tasks such as color and texture-based image retrieval (Rubner et al., 2000) and visual tracking (Schulter et al., 2017; Zhao et al., 2008; Li, 2013), EMD has demonstrated effectiveness in capturing relationships between complex structural patterns. More recently, EMD has been employed to few-shot classification tasks (Zhang et al., 2020; Xie et al., 2022a) to measure structural distances between image representations. In the context of SSL, Self-EMD (Liu et al., 2020) utilizes EMD to align dense feature embeddings in non-iconic datasets such as COCO, preserving spatial structure in feature maps to improve object detection. Unlike prior work that applies EMD in settings with rich supervised semantic information, our study introduces EMD in a fully self-supervised setting, using it to estimate the view diversity, thereby providing insights for future SSL design.

## 3 Preliminary

In this section, we provide an overview of MoCo-v2 (Chen et al., 2020b) and DINO (Caron et al., 2021), two widely used SSL frameworks that serve as the foundation of our study. Our work specifically focuses on SSLs with instance consistency (Wu et al., 2018), where each image is treated as a separate class. Both MoCo-v2 and DINO implicitly rely on this assumption, which we seek to extend to more diverse data sources to investigate the necessity of instance consistency in SSL.

MoCo-v2 (Chen et al., 2020b) employs a memory bank to store large number of negative samples, ensuring smooth updates with momentum for better consistency. It learns feature representations using the InfoNCE (Oord et al., 2018) loss:

$$\mathcal{L}_q = -\log \frac{\exp\left(q \cdot k_+/\tau\right)}{\exp\left(q \cdot k_+/\tau\right) + \sum_{k_-} \exp\left(q \cdot k_-/\tau\right)}, \tag{1}$$

where $\tau$ is the temperature, $q$ is the encoded query, $k_+$ is the positive key, and $k_-$ represents the negative keys. Note that $q$ and $k_+$ are two augmented views from the same image.

DINO (Caron et al., 2021) uses a teacher-student self-distillation framework, where the model learns categorical distributions from the `[CLS]` token of two augmented views from the same image. The teacher $\theta_t$ and the student $\theta_s$ share the same architecture, and the teacher parameters are updated with the Exponential Moving Average (EMA) of the student parameters. The knowledge is distilled from teacher $\theta_t$ to student $\theta_s$ by minimizing the cross-entropy loss:

$$\mathcal{L}_{\texttt{[CLS]}} = -P_{\theta_t}^{\texttt{[CLS]}}(v)^{\mathrm{T}} \log P_{\theta_s}^{\texttt{[CLS]}}(u), \tag{2}$$

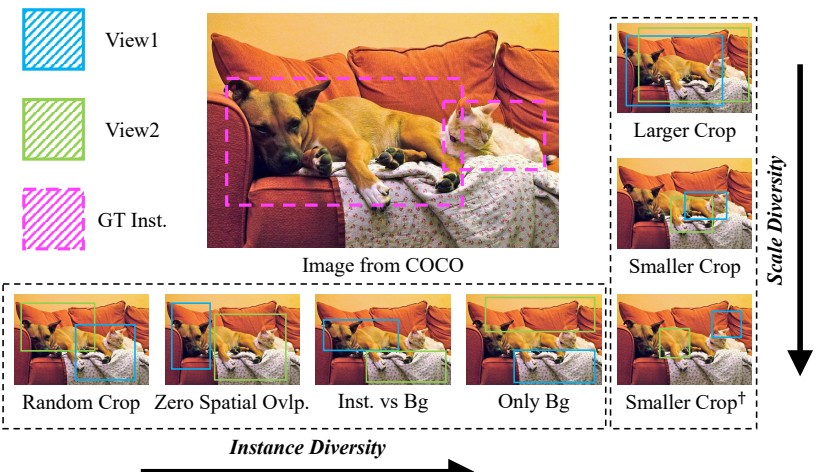

Figure 2: **Overview of positive pairs from different configurations.** This figure illustrates various positive pair configs proposed in our experiments, categorized into **Instance Diversity** and **Scale Diversity**. The **Instance Diversity** category varies the level of instance consistency between positive pairs to investigate its necessity, while the **Scale Diversity** category varies crop scales to evaluate the impact of diversity between positive pairs.

where $u$ and $v$ are two augmented views from the same image, and $P_\theta$ is the probability distribution of network $\theta$.

## 4    Delving into Instance Consistency in SSL

In this section, we conduct a comprehensive ablation study to investigate the effectiveness of SSL when instance-consistency is not guaranteed. By systematically adjusting crop configurations, we analyze whether SSL can still function effectively when positive pairs contain minimal shared instance semantics. Our findings reveal that the strict instance consistency plays a less important role in SSL than previously assumed, while view diversity plays a crucial role in enhancing SSL performance. To quantify this balance, we then introduce Earth Mover's Distance (EMD) as an estimator for quantifying diversity between augmented views, demonstrating its alignment with experimental results and its potential as a predictive measure for optimizing future SSL augmentation design. Finally, we validate our findings across diverse settings to evaluate the robustness and generalizability of SSL on a wider range of data sources. Detailed descriptions of the pre-training and downstream fine-tuning datasets, along with the experimental setups, are provided in Appendix A.

### 4.1    How Does SSL React to Different Levels of Instance Consistency?

Traditional SSL methods (Chen et al., 2020b; Caron et al., 2021; Chen et al., 2020a) are often under the instance consistency paradigm to treat each image as a separate class (Wu et al., 2018) and are pre-trained on iconic data for ensuring invariant semantic features are shared across different views of the same object. However, for non-iconic data, with complex scenes containing multiple diverse objects and varied backgrounds, random crops of the same image may contain entirely different objects or background elements, leading to the breakdown of such instance consistency, as illustrated in Figure 2. For example, two crops from an image of a pet expo might feature a cat in one view and a dog in the other. Given this, it remains unclear to what extent SSL performance is affected by different levels of instance consistency. While SSL methods like MoCo-v2 (Chen et al., 2020b) and DINO (Caron et al., 2021) can achieve strong performance on non-iconic data (Van Gansbeke et al., 2021; Mishra et al., 2022; Zhu et al., 2023), it is essential to investigate whether this holds consistently across different levels of instance consistency. This leads to our first question: **How does SSL perform under different levels of instance consistency?**

| Config | COCO | | | | | ImageNet-100 | | | | |
|---|---|---|---|---|---|---|---|---|---|---|
| | CIFAR-10 | CIFAR-100 | DTD | Pets | STL-10 | CIFAR-10 | CIFAR-100 | DTD | Pets | STL-10 |
| Baseline | 70.90 | 47.03 | 38.40 | 38.40 | 71.84 | 71.85 | 48.24 | 39.26 | 43.85 | 75.31 |
| Lower Bound | $32.72_{\ -38.18}$ | $12.40_{\ -34.63}$ | $7.29_{\ -31.11}$ | $7.28_{\ -31.12}$ | $27.59_{\ -44.25}$ | $32.64_{\ -39.12}$ | $12.22_{\ -36.02}$ | $6.65_{\ -32.61}$ | $7.09_{\ -36.76}$ | $28.36_{\ -46.95}$ |
| Spatial Ovlp. $= 0$ | $71.75_{\ +0.85}$ | $48.45_{\ +1.42}$ | $41.65_{\ +3.25}$ | $39.53_{\ +1.13}$ | $74.42_{\ +2.58}$ | $74.72_{\ +2.87}$ | $51.95_{\ +3.71}$ | $46.01_{\ +6.75}$ | $46.70_{\ +2.85}$ | $76.69_{\ +1.38}$ |
| Inst. *vs* Bg | $76.20_{\ +5.30}$ | $54.79_{\ +7.76}$ | $41.12_{\ +2.72}$ | $40.90_{\ +2.50}$ | $74.75_{\ +2.91}$ | $75.71_{\ +3.86}$ | $53.21_{\ +4.97}$ | $42.77_{\ +3.51}$ | $45.11_{\ +1.26}$ | $77.38_{\ +2.07}$ |
| Only Bg | $72.13_{\ +1.23}$ | $49.47_{\ +2.44}$ | $41.91_{\ +3.51}$ | $39.20_{\ +0.80}$ | $73.91_{\ +2.07}$ | $73.73_{\ +1.88}$ | $50.62_{\ +2.38}$ | $43.40_{\ +4.14}$ | $45.03_{\ +1.18}$ | $76.56_{\ +1.25}$ |
| Larger Crop | $67.02_{\ -3.88}$ | $43.64_{\ -3.39}$ | $33.24_{\ -5.16}$ | $29.05_{\ -9.35}$ | $69.99_{\ -1.85}$ | $66.72_{\ -5.13}$ | $42.15_{\ -6.09}$ | $34.63_{\ -4.63}$ | $36.33_{\ -7.52}$ | $72.94_{\ -2.37}$ |
| Smaller Crop | $71.36_{\ +0.46}$ | $48.28_{\ +1.25}$ | $41.76_{\ +3.36}$ | $39.81_{\ +1.41}$ | $73.69_{\ +1.85}$ | $74.97_{\ +3.12}$ | $51.59_{\ +3.35}$ | $44.63_{\ +5.37}$ | $45.90_{\ +2.05}$ | $76.92_{\ +1.61}$ |
| Smaller Crop$^{\dagger}$ | $70.34_{\ -0.56}$ | $47.26_{\ +0.23}$ | $40.43_{\ +2.03}$ | $36.44_{\ -1.96}$ | $72.26_{\ +0.42}$ | $67.72_{\ -4.13}$ | $50.21_{\ +1.97}$ | $40.96_{\ +1.70}$ | $40.37_{\ -3.48}$ | $75.65_{\ +0.34}$ |

Table 1: **Classification results with MoCo-v2 (Chen et al., 2020b) pre-trained on COCO (Lin et al., 2014) and ImageNet-100 (Deng et al., 2009).** We freeze the pre-trained weights of the SSL backbone and train a supervised linear classifier to evaluate the learned representations on five classification benchmarks (Krizhevsky et al., 2009a;b; Cimpoi et al., 2014; Parkhi et al., 2012; Coates et al., 2011). All configurations are pre-trained and linear fine-tuned for 100 epochs to ensure fair comparison. Performance gaps relative to the baseline configuration are indicated as superscripts. Smaller Crop$^{\dagger}$ denotes to **Smaller Crop with Zero Spatial Overlap** configuration.

To probe further, we empirically conduct a series of ablation experiments with MoCo-v2 and DINO pre-trained on various data sources, while systematically varying the shared instance semantics between positive pairs. We then evaluate the learned representations on classification and dense prediction tasks to assess the necessity of instance consistency on SSL performance.

**Experiment Setup.** We conduct controlled experiments to analyze the impact of different levels of instance consistency between the positive pair $(\mathbf{v_1}, \mathbf{v_2})$, obtained from augmented views[1]. Given target instances with ground-truth bounding boxes denoted as $\bigcup_{i=1}^{n} \mathbf{box}_i$, we introduce the following five primary configurations[2]:

1) ***Completely Random Crop.*** Two views $\mathbf{v_1}$ and $\mathbf{v_2}$ are randomly cropped from the same image without any spatial constraint. This configuration replicates the default setup in multi-view SSL methods (*i.e.* MoCo-v2 and DINO), which serves as the ***Baseline*** of the comparison.

2) ***Zero Spatial Overlap.*** Views are sampled with no spatial overlap to test the reliance of SSLs on the instance consistency from the shared spatial regions.

$$\text{IoU}(\mathbf{v_1}, \mathbf{v_2}) = 0$$

3) ***Instance vs Bg.*** To further reduce the instance consistency in positive pairs in the former config (two views may be partially cropped one same instance), we sample $\mathbf{v_1}$ around a foreground instance, while $\mathbf{v_2}$ contains only background information, ensuring no overlap with any instance object. Here ***Bg.*** refers to ***Background*** for simplicity.

$$\text{IoU}(\mathbf{v_1}, \mathbf{v_2}) = 0$$

$$(\exists i,\ \text{IoU}(\mathbf{v_1}, \mathbf{box}_i) > 0.8)\ \wedge\ (\forall j,\ \text{IoU}(\mathbf{v_2}, \mathbf{box}_j) < 0.1)$$

4) ***Only Bg.*** To completely remove possible instance consistency, two views are randomly sampled purely from background regions, ensuring no foreground instances are included.

$$\text{IoU}(\mathbf{v_1}, \mathbf{v_2}) = 0$$

$$\forall i,\ \text{IoU}(\mathbf{v_1}, \mathbf{box}_i) < 0.1\ \wedge\ \text{IoU}(\mathbf{v_2}, \mathbf{box}_i) < 0.1$$

5) ***Lower Bound.*** Each view is sampled from entirely different images, minimizing any possible consistency within positive pairs. This configuration serves as the lower-bound comparison to evaluate how SSL performs when no mutual information exists within positive pairs.

Detailed descriptions on the implementations of these configs are provided in Appendix A.1.

---

[1]We follow the settings from Chen et al. (2020a) to use the `RandomResizedCrop` in PyTorch with the scaling $s = (0.2, 1.0)$ and the output size of $224 \times 224$.

[2]We collectively refer to these five configurations as the **Instance Diversity** category, as illustrated in Figure 2.

| Config | COCO | | ImageNet-100 | |
| --- | --- | --- | --- | --- |
| | VOC-0712 | DOTA-v1.0 | VOC-0712 | DOTA-v1.0 |
| random init. | 53.58 | 31.59 | 53.38 | 31.59 |
| Lower Bound | 70.54 $^{-2.78}$ | 47.94 $^{-6.44}$ | 69.97 $^{-3.94}$ | 48.96 $^{-6.34}$ |
| Baseline | 73.32 | 54.38 | 73.91 | 55.30 |
| Spatial Ovlp. $= 0$ | 74.55 $^{+1.23}$ | 55.84 $^{+1.46}$ | 74.87 $^{+0.96}$ | 56.23 $^{+0.93}$ |
| Inst. *vs* Bg | 74.15 $^{+0.83}$ | 55.47 $^{+1.09}$ | 74.35 $^{+0.44}$ | 56.65 $^{+1.35}$ |
| Only Bg | 74.73 $^{+1.41}$ | 55.34 $^{+0.96}$ | 74.43 $^{+0.52}$ | 56.65 $^{+1.35}$ |
| Larger Crop | 72.14 $^{-1.18}$ | 52.09 $^{-2.29}$ | 73.40 $^{-0.51}$ | 54.06 $^{-1.24}$ |
| Smaller Crop | 74.58 $^{+1.26}$ | 55.52 $^{+1.14}$ | 74.49 $^{+0.58}$ | 56.26 $^{+0.96}$ |
| Smaller Crop$^{\dagger}$ | 73.90 $^{+0.58}$ | 54.68 $^{+0.30}$ | 74.09 $^{+0.18}$ | 55.63 $^{+0.33}$ |

Table 2: **Object detection results with MoCo-v2 (Chen et al., 2020b) pre-trained on COCO (Lin et al., 2014) and ImageNet-100 (Deng et al., 2009).** We evaluate the learned representations on VOC (Everingham et al., 2010) and DOTA (Xia et al., 2018) for object detection. All configs are pre-trained for 100 epochs for fair comparison. Random Init. refers to the backbone being randomly initialized during downstream fine-tuning.

**Results.** Table 1 presents the performance of the proposed configurations on classification tasks (Krizhevsky et al., 2009a;b; Coates et al., 2011; Parkhi et al., 2012; Cimpoi et al., 2014), while Table 2 presents results on object detection tasks (Everingham et al., 2010; Xia et al., 2018). As expected, the *Lower Bound* configuration yields the lowest performance, highlighting the importance of shared information between positive pairs for effecive SSL. Surprisingly, a notable finding is that all other configurations (***Zero Spatial Overlap***, ***Instance vs Bg***, and ***Only Bg***) outperform the baseline configuration across classification and object detection evaluations, indicating that SSL can still effectively learn representations without strict instance consistency, even surpassing the default settings.

**Discussion.** In contrast to prior beliefs (Selvaraju et al., 2021; Chuang et al., 2022), which suggests SSL should rely heavily on object-centric iconic data with strong consistent semantics in positive pairs, our findings reveal that strict instance consistency is not essential. Existing SSL methods can still learn meaningful representations when positive pairs are in the absence of strict instance consistency, as long as both views are sampled from the same image. This suggests that SSLs are capable of leveraging broader contextual cues beyond instance consistency, including shared background patterns, consistent camera viewpoints, and general color style, aligning with observations in Van Gansbeke et al. (2021). These findings highlight the potential of SSL on non-iconic data, expanding the range of a wider applicable data sources.

**Remark.** The above controlled experiments aim to investigate the necessity of the instance consistency assumption, especially when pre-training on non-iconic data, where such consistency may not naturally hold. To this end, the designed ground-truth-based view generation config is solely intended for controlled experimental setups, allowing us to systematically assess how varying levels of instance consistency between positive pairs affect SSL performance. These configs are not designed to serve as data augmentation pipelines for practical SSL pre-training use.

Additionally, certain configs, like ***Zero Spatial Overlap***, may not fully eliminate instance consistency, as non-overlapping regions of a large object instance may still share the same semantic information. However, we would like to emphasize that these configs only serve to stepwise reduce the degree of semantic consistency, achieving by firstly removing the spatial shared overlapping content between augmented views. In practice, it offers a controlled setup to study the impact of different levels of instance consistency on SSL performance.

> **Takeaway 1**
>
> In contrast to prior beliefs of instance consistency (Selvaraju et al., 2021; Chuang et al., 2022), our experiments empirically show that SSL can learn meaningful representations even when positive pairs contain minimal shared instance semantics.

### 4.2 How Much View Diversity is Beneficial?

As explored in Section 4.1, instance consistency appears to be less critical for effective SSL. Meanwhile, while our configurations reduce the instance consistency, they simultaneously increase diversity and reduce redundancy between positive pairs, yet still serve as a valid, and even better supervision signal. This raises a new question: **How much diversity between positive pairs is beneficial, and when does it become detrimental?** This aligns with findings from Tian et al. (2020), which suggest that higher diversity between views can enhance SSL performance.

To further validate this hypothesis, we conduct a set of orthogonal experiments focusing on the diversity between positive pairs, specifically by varying crop scales. According to the Law of Large Numbers, smaller crop scales naturally reduce the likelihood of overlapping regions between views, while larger crops increase spatial redundancy. Additionally, smaller crops inherently capture less information per view, potentially increasing the diversity within positive pairs. The following experiments evaluate how different levels of view diversity impact SSL representation learning.

**Experiment Setup.** We introduce three primary configurations[3], which complement the previous configs in Section 4.1 to regulate the diversity between positive pairs by systematically varying crop scales.

1) ***Smaller Crop.*** This configuration applies a smaller crop scale to reduce the area captured by each view. The smaller region minimizes shared information to increase diversity between positive pairs. .

2) ***Larger Crop.*** Larger crop scales increase the area captured by each view, creating larger overlap to preserve more shared information, thereby reducing the diversity.

3) ***Smaller Crop with Zero Spatial Overlap.*** To maximize the diversity, this configuration combines the smaller crop with the zero spatial overlap constraint, ensuring no overlapping spatial overlapping within positive pairs. This setting enforces the lowest shared information, allowing us to evaluate SSL's ability to learn from highly diverse positive pairs.

All configurations maintain a consistent output size of $224 \times 224$, aligning with the settings in Section 4.1. Detailed descriptions and ablation studies on the selection of crop scales are provided in Appendix A.1.

**Results.** Table 1 presents the classification results of varying crop scales, and Table 2 presents the detection results. The findings confirm our hypothesis that increasing diversity between positive pairs in SSL can enhance downstream performance: configurations with higher diversity (*i.e.* ***Smaller Crop***) consistently outperform the baseline. Conversely, the ***Larger Crop*** configuration, which reduces diversity by preserving more shared information, leads to a significant performance drop, suggesting that excessive redundancy between views can hinder SSL effectiveness. Interestingly, while ***Smaller Crop*** and ***Zero Spatial Overlap*** individually boost performance, their combination, ***Smaller Crop with Zero Spatial Overlap***, does not yield additional gains and instead results in a slight performance drop.

**Discussion.** These findings highlight the importance of striking a balance in view diversity for effective SSL. Increasing view diversity through configurations like, ***Smaller Crop*** and ***Zero Spatial Overlap***, effectively reduces mutual information, encouraging the model to discover more fine-grained visual consistencies, thus enhancing SSL performance. However, the observed performance drop when combining these two configs suggests that excessive diversity can be detrimental, as minimizing shared information beyond a certain threshold hinders the model's ability to learn meaningful representations.

This aligns with prior study (Tian et al., 2020) that describes a U-shaped relationship between mutual information and downstream performance – indicating that while reducing redundancy is beneficial, completely eliminating shared information can degrade SSL's effectiveness. These insights imply the existence of an optimal range for view diversity, while mutual information is minimized yet remains sufficient for effective SSL learning. This highlights the need for a quantitative estimator to evaluate and balance shared information between views, guiding the augmentation process toward optimal performance, which is to be elaborated in Section 4.3.

---

[3]We collectively refer to these three configurations as the **Scale Diversity** category, as illustrated in Figure 2.

---

**Takeaway 2**

Our findings empirically show that increasing the diversity in positive pairs encourages the discovery of more fine-grained visual consistencies in SSL, thus enhancing downstream performance. However, excessive diversity may degrade SSL's effectiveness.

---

### 4.3 Earth Mover's Distance as Diversity Estimator

Experiments in Sections 4.1 and 4.2 show that view diversity between positive pairs plays a crucial role in SSL performance. To quantify this diversity and give an estimation of the effectiveness of different view augmentations, we adopt Earth Mover's Distance (EMD) as an estimator to measure the shared information between positive pairs. By evaluating the similarity between augmented views within positive pairs, EMD provides a robust estimation of augmentation quality before model pre-training. This enables a systematic approach to optimize the positive pair selection for improving SSL effectiveness.

**Background.** To accurately measure the distance or the similarity between two views, we require a metric that account for spatial variations in the possible data sources of SSL. To accommodate non-iconic data containing, where multiple objects or complex scene environments appear across different views, the simple L2 distance metric is unsuitable due to its reliance on strict spatial alignment. Furthermore, previous experiments in Section 4.1 show that SSL can perform effective feature learning without strict instance consistency, suggesting that the used view similarity should not simply focus on the pixel or the image level, which needs to further explore in the feature space. Therefore, we adopt Earth Mover's Distance (EMD) to automatically identify correspondences between views based on their visual features.

Earth Mover's Distance (Rubner et al., 2000; Zhang et al., 2020) quantifies the distance between two distributions by computing the minimum cost needed to transform one distribution into another, making it a well-established formulation of the optimal transport problem (OTP). In our case, EMD measures the distance between the given feature maps of two augmented views $\mathbf{X}, \mathbf{Y} \in \mathcal{R}^{N \times D}$, where $N$ denotes the number of feature vectors in each feature map and $D$ represents the feature dimension. More details on the definition and computation of EMD are provided in Appendix A.3.

**Implementations.** As shown in Figure 3, to compute the EMD-based similarity score, we follow the settings in Zhang et al. (2020) to employ two strategies for generating feature vectors from two views in the positive pair. In both strategies, we first generate two augmented views of each image and extract their features using a pre-trained ResNet-50 (He et al., 2016). To account for potential scale differences between augmented views, each strategy uses distinct cropping patterns:

1) *Grid-based*: Each view is divided into uniform grids, with grid factors of 2 and 3. Each grid cell serves as a separate patch, which is then passed through a pre-trained model to generate the feature vector.

2) *Sampling-based*: Each view is randomly sampled into 9 patches, varying the sizes and aspect ratios to introduce scale diversity. Each sampled patch is resized with the input size of 84 before being processed by the pre-trained model to produce its corresponding feature vector.

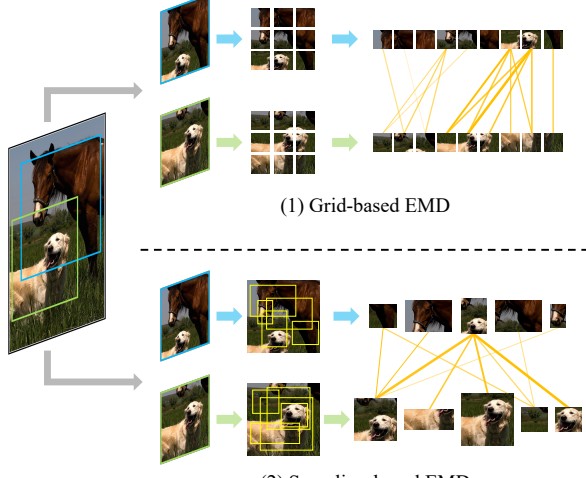

(1) Grid-based EMD

(2) Sampling-based EMD

Figure 3: **Two strategies for EMD-based similarity score.** Different line widths between patches represent the mass transported under the OT plan used in EMD computation. Thicker lines correspond to higher transport mass, indicating stronger correspondence between the two patches.

**Results.**   To validate the effectiveness of Earth Mover's Distance (EMD) as an estimator for assessing similarity between augmented views, we compute the EMD similarity score for all the proposed configurations in Sections 4.1 and 4.2. Figure 4 reveals a clear reverse-U relationship between the EMD score and downstream accuracy, evaluated using the two proposed cropping strategies. Configurations with moderate EMD scores[4] (ranging from 3 to 4) consistently yield the highest performance. This suggests that when the diversity between positive pairs is within an optimal range, the shared mutual information between views remains sufficient for effective feature learning, leading to improved downstream performance. In contrast, configurations with either very high (above 5) or very low EMD scores (below 2) exhibit a drop in downstream accuracy, indicating that extreme overlap or excessive diversity between views can hinder SSL's ability to learn meaningful feature representations. Plots for SSL pre-trained on ImageNet (Deng et al., 2009) are provided in Appendix B.5.

**Discussion.**   Our analysis demonstrates that Earth Mover's Distance (EMD) is an effective estimator of view diversity in SSL, providing an approach to quantify the mutual information between augmented views. By leveraging EMD, we can evaluate and regulate view diversity to ensure it remains within an optimal range for effective SSL training. This insight suggests that EMD can serve as a valuable measure for guiding augmentation strategies in future SSL framework design, allowing the prediction of the effectiveness of positive pair selection to enhance downstream performance.

**Suggestions for Positive Pair Selection.**   Our experimental results suggest a potential improvement for positive pair selection in SSL: pre-calculating EMD scores before pre-training. Specifically, our findings indicate that the optimal EMD range lies between the baseline score (upper bound) and that of Smaller Crop$^{\dagger}$ (lower bound), where the latter can also represent positive pairs with excessive diversity. This insight offers a promising alternative to the conventional random cropping approach in SSL. Our experiment results with **Zero Spatial Overlap** and **Smaller Crop** configurations also support the effectiveness of this strategy.

> **Takeaway 3**
>
> Our findings reveal that Earth Mover's Distance (EMD) effectively quantifies mutual information between positive pairs, making it a predictive measure of view diversity for effective SSL.

### 4.4   Validation Across Diverse Settings

We have demonstrated the generalizability of our findings on both iconic and non-iconic data, validating their effectiveness on both classification and object detection tasks. Building on these results, we provide additional evaluations to further assess the robustness of our insights across diverse application scenarios.

**Diverse SSL Methods.**   To ensure our findings are not limited to contrastive learning, we further expand our analysis to the DINO (Caron et al., 2021) framework as shown in Figures 4c and 4d. These results demonstrate that our insights apply beyond contrastive-based methods, generalizing to a broader range of instance consistency-based SSL approaches. Numerical results and further analysis are provided in Appendix B.1.

**Diverse Tasks & Experimental Settings.**   We further validate our findings by extending experiments to more downstream tasks, including instance segmentation and depth prediction, assessing the generalizability of our insights across different learning objectives. Additionally, we examine various experimental settings, such as frozen-backbone tuning, extensive dataset transfer scenarios, and extended pre-training durations, to comprehensively evaluate our findings under diverse training conditions. Detailed results and analysis are provided in Appendices B.2 and B.3.

> **Takeaway 4**
>
> Our validation experiments confirm the generalizability of our findings across a range of settings, while highlighting the adaptability of the proposed EMD diversity estimator on various data sources.

---

[4]EMD scores in Figure 4 are scaled by 10 for better visualization.

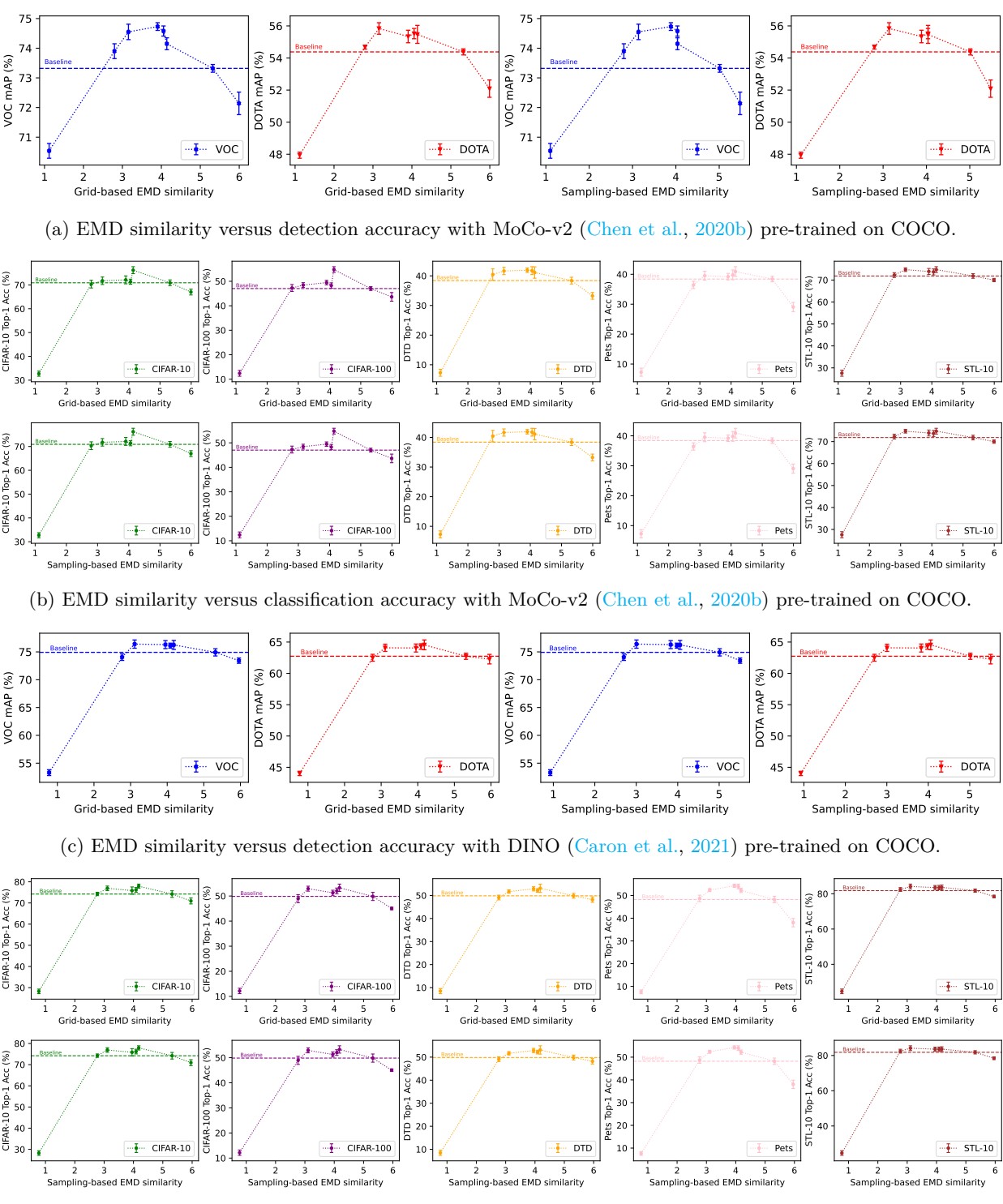

(a) EMD similarity versus detection accuracy with MoCo-v2 (Chen et al., 2020b) pre-trained on COCO.

(b) EMD similarity versus classification accuracy with MoCo-v2 (Chen et al., 2020b) pre-trained on COCO.

(c) EMD similarity versus detection accuracy with DINO (Caron et al., 2021) pre-trained on COCO.

(d) EMD similarity versus classification accuracy with DINO (Caron et al., 2021) pre-trained on COCO.

Figure 4: **EMD similarity versus detection and classification accuracy.** The similarity scores between views are plotted against object detection results in (a), (c) and classification results in (b), (d). Baseline configuration is highlighted for reference. EMD scores are scaled by a factor of 10 for better visualization. Across all settings, the results exhibit a clear reverse-U curve, supporting the hypothesis that an optimal range of view diversity exists for effective SSL.

## 5   Conclusion

In this paper, we investigate a critical research question: **Is instance consistency a strict requirement for self-supervised learning (SSL)?** To explore this, we systematically analyze the effectiveness of SSL when instance consistency is not guaranteed. Our findings reveal that SSL can still learn meaningful representations even when positive pairs contain minimal shared instance semantics, suggesting that strict instance consistency is not essential for effective SSL learning. Furthermore, our analysis further reveals that increasing diversity between positive pairs, such as enforcing zero overlapping or using smaller crop scales, can enhance performance across various downstream tasks. However, excessive diversity is found to reduce effectiveness, indicating the existence of an optimal range for view diversity. To quantify this diversity, we adopt Earth Mover's Distance (EMD) as a metric to measure mutual information between views, finding that moderate EMD values correlate with improved SSL learning, providing a useful estimator for guiding positive pair selection in future SSL framework design. We validate our findings across a range of settings, highlighting the practical applicability of our insights for effective SSL across diverse application scenarios.

**Broader Impacts.**   While our findings highlight the flexibility of SSL without strict instance consistency, particularly on non-iconic data, it may also lead to unreliable representations if applied in sensitive domains such as healthcare or autonomous driving without proper validation. Furthermore, training on large-scale, unfiltered Internet data may amplify societal and demographic biases due to under-representation or skewed content. We therefore recommend domain-specific validation and bias auditing when applying our findings to real-world applications.

**Acknowledgements.**   This research is supported by the EDB Space Technology Development Programme under Project S22-19016-STDP and the National Natural Science Foundation of China under Grant 62576089. The authors sincerely thank the Reviewers and Action Editor for their valuable and constructive feedback.

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

# Contents

# A    Implementation Details

## A.1    Pre-training Setup

**Dataset.**   We conduct SSL pre-training on two datasets: COCO for non-iconic data and ImageNet-100 for object-centric data. COCO (Lin et al., 2014) is a large non-iconic dataset with 118k training images containing approximately 896k labeled objects, averaging 7 objects per image. In contrast, ImageNet-100 is a subset of the object-centric dataset ImageNet-1K (Deng et al., 2009), consisting of 100 randomly selected classes, with 128k images in total. The selection ensures alignment with the number of training samples in COCO for fair comparisons. The specific ImageNet-100 classes used in our experiments are listed in Table A. All images of both datasets are used for SSL pre-training in our experiments.

| List of ImageNet-100 classes | | | | |
| --- | --- | --- | --- | --- |
| n01443537 | n01484850 | n01514668 | n01518878 | n01531178 |
| n01532829 | n01537544 | n01580077 | n01582220 | n01601694 |
| n01608432 | n01632458 | n01665541 | n01669191 | n01704323 |
| n01728920 | n01729977 | n01755581 | n01756291 | n01797886 |
| n01807496 | n01824575 | n01843065 | n01847000 | n01871265 |
| n01872401 | n01873310 | n01950731 | n01968897 | n01978287 |
| n01983481 | n01985128 | n02002724 | n02009912 | n02011460 |
| n02017213 | n02018207 | n02018795 | n02088364 | n02088632 |
| n02090622 | n02090721 | n02091032 | n02091467 | n02092002 |
| n02093859 | n02096437 | n02097047 | n02097209 | n02100236 |
| n02101388 | n02105855 | n02110627 | n02110806 | n02113624 |
| n02113978 | n02114548 | n02114855 | n02116738 | n02130308 |
| n02137549 | n02165105 | n02174001 | n02177972 | n02281787 |
| n02319095 | n02364673 | n02415577 | n02417914 | n02442845 |
| n02443114 | n02444819 | n02447366 | n02480495 | n02481823 |
| n02493793 | n02640242 | n02643566 | n02655020 | n02727426 |
| n02776631 | n02782093 | n02797295 | n02804414 | n02823428 |
| n02834397 | n02865351 | n02869837 | n02871525 | n02877765 |
| n02883205 | n02917067 | n02927161 | n02939185 | n02948072 |
| n02965783 | n02966687 | n02977058 | n02992529 | n02999410 |

Table A: **List of classes from ImageNet-100.**    These classes are randomly sampled from the original ImageNet-1K dataset (Deng et al., 2009).

**Setup.**   All models are pre-trained from scratch for 100 epochs. For the backbone, we use ResNet-50 (He et al., 2016) in MoCo-v2 (Chen et al., 2020b), and ViT-S (Dosovitskiy et al., 2021) with the patch size of 16 in DINO (Caron et al., 2021). Specifically, for MoCo-v2, we set the batch size as 256 and the learning rate as 0.3 with the SGD optimizer. For DINO, we set the batch size as 256 and the learning rate as 0.0005 with the AdamW optimizer. All other training hyper-parameters follow the original settings in their respective implementations. All pre-training experiments are conducted on NVIDIA RTX A6000 GPUs.

**Implementations.**   We provide the implementation details for the proposed configs in our ablation experiments.

For the **Instance Diversity** category, we need to utilize the locations of object instances in the given image. We use the GT annotations provided in COCO as the reference to obtain this object instance information. However, unlike COCO, which includes GT bounding-box annotations, we need to self-identify the locations of foreground instances in ImageNet-100. We adopt two unsupervised approaches to generate pseudo masks for object instances: MaskCut (Wang et al., 2023), and Selective Search (Uijlings et al., 2013).

MaskCut (MC) is introduced in CutLER (Wang et al., 2023), which combines Normalized Cuts (NCut) (Shi & Malik, 2000) and Conditional Random Fields (CRFs) (Krähenbühl & Koltun, 2011) to discover multiple object instance masks without any supervision. We adopt the official implementation of MaskCut to obtain pseudo masks for ImageNet-100. Selective Search (SS) (Uijlings et al., 2013) is a classic unsupervised object proposal generation method, which leverages color similarity, texture similarity, region size, and fit between regions to identify object candidates. We use the object proposals generated by SoCo's (Wei et al., 2021)

official implementation. To reduce noise and exclude tiny instances, we limit the maximum number of objects to 3 per image for both approaches. The pseudo masks generated by these methods are used to implement the configs in ablation experiments in the **Instance Diversity** category. A discussion of this two approaches is provided in Appendix B.4.

For the **Scale Diversity** category, we carefully adjust crop scales within the pre-training dataset. For COCO, we use the average object instance size derived from dataset annotations, resulting in a scaling range of $s = (0.08, 0.4)$ for ***Smaller Crop***. Meanwhile, we apply a scaling range of $s = (0.4, 1.0)$, which doubles the scale in the default setting for ***Larger Crop***. Considering the object scale difference in two datasets, for ImageNet-100, we apply a scaling range of $s = (0.18, 0.9)$ for ***Smaller Crop***, and $s = (0.4, 1.0)$ for ***Larger Crop***. The crop scales are selected based on the ablation studies in Appendix B.4.

## A.2 Downstream Fine-tuning Setup

**Dataset.** We evaluate the pre-trained models on a board range of downstream evaluation tasks including classification, object detection, instance segmentation and depth prediction. For object detection, we use PASCAL VOC-0712 (Everingham et al., 2010) for general object detection, and DOTA-v1.0 (Xia et al., 2018) for aerial object detection. For classification, we utilize five small-scale classification datasets: CIFAR-10 (Krizhevsky et al., 2009a), CIFAR-100 (Krizhevsky et al., 2009b), DTD (Cimpoi et al., 2014), Oxford Pets (Parkhi et al., 2012), and STL-10 (Coates et al., 2011). Additionally, COCO (Lin et al., 2014) is included for the in-distribution evaluation on object detection and instance segmentation tasks. We also include depth prediction on NYUd (Silberman et al., 2012) to demonstrate the generalizability of our findings to 3D downstream tasks.

**Setup.** All downstream experiments are conducted on NVIDIA RTX A6000 GPUs. We list the setups for downstream tasks as follows:

- **Object Detection**: Evaluations are performed using MMDetection (Chen et al., 2019) and MMRotate (Zhou et al., 2022). Specifically, Faster R-CNN (Ren et al., 2015) with the $24k$ iteration schedule is used for PASCAL VOC general object detection. Oriented R-CNN (Xie et al., 2021) with the $1\times$ schedule is used for DOTA aerial object detection, and Mask R-CNN (He et al., 2017) with the $1\times$ schedule is used for COCO object detection and instance segmentation. The batch size is set to 2 per GPU, with other hyper-parameters following the default settings. We report the mean Average Precision (mAP) as the evaluation metric for object detection.

- **Classification**: We freeze the pre-trained SSL backbone and train a supervised linear classifier on top of it to perform the classification evaluation. For MoCo-v2, we follow the evaluation protocol in Peng et al. (2022), the linear classifier is trained for 100 epochs with an initial learning rate of 10.0, reduced by a factor of 0.1 at the $60th$ and $80th$ epochs. For DINO, we follow Caron et al. (2021) to train the linear classifier for 100 epochs with an initial learning rate of 0.001, optimizing by SGD using a cosine annealing schedule. The batch size is set to 512 across all five datasets for both cases. We report the Top-1 Accuracy as the evaluation metric for classification.

- **Depth Prediction**: Evaluations are performed using Monocular-Depth-Estimation-Toolbox (Li, 2022) based on MMSegmentation (Contributors, 2020). Adabins (Bhat et al., 2021) with $2\times$ schedule is used for NYUd depth prediction. The batch size is set to 8 per GPU, with other hyper-parameters following the default settings. We report the RMSE as the evaluation metric for depth prediction.

## A.3 EMD-based Similarity Score Computation

We provide more details of the definition and computation of EMD-based similarity below.

**Definition.** Earth Mover's Distance (Rubner et al., 2000; Zhang et al., 2020) quantifies the distance between two distributions by computing the minimum cost needed to transform one distribution into another, which has the form of the well-studied optimal transport problem (OTP). In our case, EMD measures the distance between the given feature maps of the two augmented views $\mathbf{X}, \mathbf{Y} \in \mathcal{R}^{N \times D}$, where $N$ denotes the number of

---

**Algorithm A:** Procedure for EMD-based Similarity Score

---

**Data:** Two augmented views $\mathbf{X}, \mathbf{Y} \in \mathcal{R}^{N \times D}$.
**Result:** Similarity score $S(\mathbf{X}, \mathbf{Y})$.

**1** Flatten feature maps into local feature representations
$\mathcal{X}, \mathcal{Y} \leftarrow \{\mathbf{x}_i\}_{i=1}^N, \{\mathbf{y}_j\}_{j=1}^N$ ;

**2** Define the overall transportation polytope
$U(s, d) := \{P \in \mathcal{R}_+^{N \times N} | P\mathbb{1} = \mathbf{s}, P^T\mathbb{1} = \mathbf{d}\}$;

**3** Compute the cost matrix
$C_{ij} \leftarrow 1 - \frac{\mathbf{x}_i^T \mathbf{y}_j}{\|\mathbf{x}_i\|\|\mathbf{y}_j\|}$;

**4** Solve optimal transport via *Sinkhorn-Knopp* iteration
$P^* = \arg\min_{P \in U(s,d)} \langle P, C \rangle - \frac{1}{\lambda}h(P)$;

**5** Compute the similarity score
$S(\mathbf{X}, \mathbf{Y}) \leftarrow \langle P, 1 - C \rangle$;

---

vectors in each feature map and $D$ is the feature dimension. We first flatten these maps into two sets of local feature representations $\mathcal{X} = \{\mathbf{x}_i | i = 1, 2, ..., N\}$ and $\mathcal{Y} = \{\mathbf{y}_j | j = 1, 2, ..., N\}$, where each $\mathbf{x}_i$ and $\mathbf{y}_j$ represents a local vector at a specific spatial location in the given view.

The EMD between these two feature maps is then defined as the minimum "transport cost" required to transfer units from "suppliers" in $\mathcal{X}$ to "demanders" in $\mathcal{Y}$, where each supplier $\mathbf{x}_i$ has $s_i$ units to transport, and each demander $\mathbf{y}_j$ requires $d_j$ units. The roles of suppliers and demanders can be switched without affecting the total transportation cost. The overall transportation polytope can be formulated as follows:

$$U(s, d) := \{P \in \mathcal{R}_+^{N \times N} | P\mathbb{1} = \mathbf{s}, P^T\mathbb{1} = \mathbf{d}\}. \tag{A}$$

Here $\mathbb{1} \in \mathcal{R}^N$ represents the all-ones vectors, while $\mathbf{s}$ and $\mathbf{d}$ are vectorized forms of $\{s_i\}$ and $\{d_j\}$, respectively. These vectors are also referred to as the marginal weights of matrix $P$ across its rows and columns. We then define the cost matrix $C_{ij}$ to represent the cost per unit transported from supplier node $\mathbf{x}_i$ to demander node $\mathbf{y}_j$ according to their cosine distances as:

$$C_{ij} = 1 - \frac{\mathbf{x}_i^T \mathbf{y}_j}{\|\mathbf{x}_i\|\|\mathbf{y}_j\|}, \tag{B}$$

With this notation, we can define the EMD as:

$$\text{OT}(s, d) := \min_{P \in U(s,d)} \langle P, C \rangle, \tag{C}$$

where $\text{OT}(s, d)$ is the total transportation cost and $\langle \cdot, \cdot \rangle$ denotes the Frobenius dot product of two matrices.

**Computation Details.** To find the optimal assignment matrix $P^*$, we consider the OTP as a Linear Programming problem by using *Sinkhorn-Knopp* iteration (Sinkhorn & Knopp, 1967; Cuturi, 2013), which introduces a entropy constraint term $h$:

$$P^* = \arg\min_{P \in U(s,d)} \langle P, C \rangle - \frac{1}{\lambda}h(P), \tag{D}$$

where $h(P)$ is the regularization of the entropy of the assignments, and $\lambda$ is a constant hyper-parameter to control the intensity of regularization term. After repeating $T$ times iterations ($T = 10$ in our case), the approximate optimal assignment $P^*$ can be obtained.

For our case, high values of $P_{ij}^*$ indicate a low transport cost from $\mathbf{x}_i$ to $\mathbf{y}_j$, allowing maximum unit transfer, which suggests that $\mathbf{x}_i$ and $\mathbf{y}_j$ have similar features, thus potentially sharing more meaningful mutual information. Therefore, we can compute the similarity score $S$ between the feature representations within two augmented views as:

$$S(\mathbf{X}, \mathbf{Y}) = \langle P, 1 - C \rangle, \tag{E}$$

| Config | COCO | | | | | ImageNet-100 | | | | |
|---|---|---|---|---|---|---|---|---|---|---|
| | CIFAR-10 | CIFAR-100 | DTD | Pets | STL-10 | CIFAR-10 | CIFAR-100 | DTD | Pets | STL-10 |
| Baseline | 74.21 | 49.84 | 49.84 | 48.21 | 81.79 | 77.54 | 53.88 | 53.99 | 57.57 | 84.60 |
| Lower Bound | 28.25 $^{-45.96}$ | 12.16 $^{-37.68}$ | 8.49 $^{-41.35}$ | 7.63 $^{-40.58}$ | 24.65 $^{-57.14}$ | 28.68 $^{-48.86}$ | 11.07 $^{-42.81}$ | 6.49 $^{-47.50}$ | 8.77 $^{-48.80}$ | 26.00 $^{-58.60}$ |
| Spatial Ovlp. $= 0$ | 76.92 $^{+2.71}$ | 52.92 $^{+3.08}$ | 51.67 $^{+1.83}$ | 52.38 $^{+4.17}$ | 84.14 $^{+2.35}$ | 78.57 $^{+1.03}$ | 55.03 $^{+1.15}$ | 57.14 $^{+3.15}$ | 59.14 $^{+1.57}$ | 86.56 $^{+1.96}$ |
| Inst. $vs$ Bg | 77.92 $^{+3.71}$ | 53.28 $^{+3.44}$ | 53.12 $^{+3.28}$ | 52.22 $^{+4.01}$ | 83.62 $^{+1.83}$ | 80.24 $^{+2.70}$ | 56.20 $^{+2.32}$ | 57.87 $^{+3.88}$ | 64.05 $^{+6.48}$ | 86.01 $^{+1.41}$ |
| Only Bg | 75.85 $^{+1.64}$ | 51.28 $^{+1.44}$ | 52.88 $^{+3.04}$ | 54.26 $^{+6.05}$ | 83.56 $^{+1.77}$ | 79.73 $^{+2.19}$ | 56.86 $^{+2.98}$ | 56.78 $^{+2.79}$ | 61.22 $^{+3.65}$ | 86.36 $^{+1.76}$ |
| Larger Crop | 70.95 $^{-3.26}$ | 45.03 $^{-4.81}$ | 48.19 $^{-1.65}$ | 38.05 $^{-10.16}$ | 78.44 $^{-3.35}$ | 75.31 $^{-2.23}$ | 49.43 $^{-4.45}$ | 52.29 $^{-1.70}$ | 53.88 $^{-3.69}$ | 81.24 $^{-3.36}$ |
| Smaller Crop | 76.19 $^{+1.98}$ | 52.04 $^{+2.20}$ | 52.19 $^{+2.35}$ | 53.96 $^{+5.75}$ | 83.51 $^{+1.72}$ | 79.84 $^{+2.30}$ | 55.34 $^{+1.46}$ | 58.35 $^{+4.36}$ | 66.72 $^{+9.15}$ | 86.91 $^{+2.31}$ |
| Smaller Crop$^{\dagger}$ | 74.26 $^{+0.05}$ | 48.94 $^{-0.90}$ | 49.10 $^{-0.74}$ | 48.73 $^{+0.52}$ | 82.44 $^{+0.65}$ | 76.51 $^{-1.03}$ | 52.82 $^{-1.06}$ | 52.98 $^{-1.01}$ | 56.58 $^{-0.99}$ | 84.09 $^{-0.51}$ |

Table B: **Classification results with DINO (Caron et al., 2021) pre-trained on COCO (Lin et al., 2014) and ImageNet-100 (Deng et al., 2009).** We freeze the pre-trained weights of the SSL backbone and train a supervised linear classifier to evaluate the learned representations on five classification benchmarks (Krizhevsky et al., 2009a;b; Cimpoi et al., 2014; Parkhi et al., 2012; Coates et al., 2011). All configurations are pre-trained and linear fine-tuned for 100 epochs to ensure fair comparison. Performance gaps relative to the baseline configuration are indicated as superscripts. Smaller Crop$^{\dagger}$ denotes to **_Smaller Crop with Zero Spatial Overlap_** configuration.

| Config | COCO | | ImageNet-100 | |
|---|---|---|---|---|
| | VOC-0712 | DOTA-v1.0 | VOC-0712 | DOTA-v1.0 |
| Random Init. | 58.62 | 46.96 | 58.62 | 46.96 |
| Lower Bound | 53.30 $^{-21.62}$ | 44.03 $^{-18.70}$ | 53.28 $^{-22.32}$ | 44.27 $^{-19.76}$ |
| Baseline | 74.92 | 62.73 | 75.60 | 64.03 |
| Spatial Ovlp. $= 0$ | 76.40 $^{+1.48}$ | 64.08 $^{+1.35}$ | 77.06 $^{+1.46}$ | 66.98 $^{+2.95}$ |
| Inst. $vs$ Bg | 76.25 $^{+1.33}$ | 64.55 $^{+1.82}$ | 77.80 $^{+2.20}$ | 66.49 $^{+2.46}$ |
| Only Bg | 76.31 $^{+1.39}$ | 64.05 $^{+1.32}$ | 76.76 $^{+1.16}$ | 65.45 $^{+1.42}$ |
| Larger Crop | 73.43 $^{-1.49}$ | 62.27 $^{-0.46}$ | 74.45 $^{-1.15}$ | 63.18 $^{-0.85}$ |
| Smaller Crop | 76.12 $^{+1.20}$ | 64.29 $^{+1.56}$ | 76.81 $^{+1.21}$ | 65.72 $^{+1.69}$ |
| Smaller Crop$^{\dagger}$ | 74.06 $^{-0.86}$ | 62.49 $^{-0.24}$ | 75.35 $^{-0.25}$ | 64.31 $^{+0.28}$ |

Table C: **Object detection results with DINO (Caron et al., 2021) pre-trained on COCO (Lin et al., 2014) and ImageNet-100 (Deng et al., 2009).** We evaluate the learned representations on VOC (Everingham et al., 2010) and DOTA (Xia et al., 2018) for object detection. All configs are pre-trained for 100 epochs for fair comparison. Random Init. refers to the backbone being randomly initialized during downstream fine-tuning.

where $1 - C$ denotes the cosine similarity between two local feature vectors.

We provide the pseudo code for computing the Earth Mover's Distance (EMD)-based similarity score between two augmented views in Algorithm A.

# B  Additional Experiment Results

In this section, we present additional experiment results under diverse settings to further validate the generalizability of our findings.

## B.1  Validation Across Diverse SSL Methods

To assess the broader applicability of our findings, we extend the ablation experiments on the effectiveness of instance consistency to another SSL framework, DINO (Caron et al., 2021). Unlike contrastive learning-based methods MoCo-v2 (Chen et al., 2020b), DINO employs a self-distillation approach while still relying on instance consistency during knowledge distillation, which treats different views of the same image as positive pairs. For fair comparison, we adopt the same experiment setup as MoCo-v2: pre-training on COCO / ImageNet-100 dataset and fine-tuning for classification and object detection evaluations.

**Results.**  As shown in Tables B and C, results on DINO align closely with those observed on MoCo-v2: increasing diversity between positive pairs consistently enhances baseline performance, while excessive diversity

| Config | Pre-trained | COCO Object Detection | | | COCO Instance Segmentation | | |
|---|---|---|---|---|---|---|---|
| | | $AP^b$ | $AP^b_{50}$ | $AP^b_{75}$ | $AP^m$ | $AP^m_{50}$ | $AP^m_{75}$ |
| Baseline | COCO | 34.62 | 52.95 | 37.39 | 31.28 | 50.10 | 33.33 |
| Spatial Ovlp. $= 0$ | COCO | 35.19 $^{+0.57}$ | 53.51 $^{+0.56}$ | 38.35 $^{+0.96}$ | 31.82 $^{+0.54}$ | 50.82 $^{+0.72}$ | 34.19 $^{+0.86}$ |
| Smaller Crop | COCO | 35.07 $^{+0.45}$ | 53.31 $^{+0.36}$ | 38.09 $^{+0.70}$ | 31.71 $^{+0.43}$ | 50.52 $^{+0.42}$ | 33.98 $^{+0.65}$ |
| Smaller Crop$^†$ | COCO | 34.78 $^{+0.16}$ | 53.05 $^{+0.10}$ | 37.87 $^{+0.48}$ | 31.46 $^{+0.18}$ | 50.28 $^{+0.18}$ | 33.76 $^{+0.43}$ |
| Baseline | ImageNet-100 | 34.80 | 53.29 | 37.71 | 31.59 | 50.53 | 33.95 |
| Spatial Ovlp. $= 0$ | ImageNet-100 | 35.09 $^{+0.29}$ | 53.56 $^{+0.27}$ | 38.12 $^{+0.41}$ | 32.00 $^{+0.41}$ | 51.01 $^{+0.48}$ | 34.28 $^{+0.33}$ |
| Smaller Crop | ImageNet-100 | 35.08 $^{+0.28}$ | 53.53 $^{+0.24}$ | 38.05 $^{+0.34}$ | 31.80 $^{+0.21}$ | 50.74 $^{+0.21}$ | 34.11 $^{+0.16}$ |
| Smaller Crop$^†$ | ImageNet-100 | 34.87 $^{+0.07}$ | 53.33 $^{+0.04}$ | 37.87 $^{+0.16}$ | 31.66 $^{+0.07}$ | 50.55 $^{+0.02}$ | 34.05 $^{+0.10}$ |

Table D: **Object detection and instance segmentation results with MoCo-v2 (Chen et al., 2020b) pre-trained on COCO (Lin et al., 2014) and ImageNet-100 (Deng et al., 2009).** We evaluate the learned representations on COCO (Lin et al., 2014) for object detection and instance segmentation.

| Config | COCO | ImageNet-100 |
|---|---|---|
| | NYUd RMSE $\downarrow$ | |
| Random Init. | 0.7467 | 0.7467 |
| Lower Bound | 0.6564 $^{+0.1081}$ | 0.6859 $^{+0.1357}$ |
| Baseline | 0.5483 | 0.5502 |
| Spatial Ovlp. $= 0$ | 0.5154 $^{-0.0329}$ | 0.5221 $^{-0.0281}$ |
| Inst. $vs$ Bg | 0.5149 $^{-0.0334}$ | 0.5157 $^{-0.0345}$ |
| Only Bg | 0.5183 $^{-0.0300}$ | 0.5189 $^{-0.0313}$ |
| Larger Crop | 0.5626 $^{+0.0143}$ | 0.5593 $^{+0.0091}$ |
| Smaller Crop | 0.5199 $^{-0.0284}$ | 0.5114 $^{-0.0388}$ |
| Smaller Crop$^†$ | 0.5596 $^{+0.0113}$ | 0.5571 $^{+0.0069}$ |

Table E: **Depth prediction results with MoCo-v2 (Chen et al., 2020b) pre-trained on COCO (Lin et al., 2014) and ImageNet-100 (Deng et al., 2009).** We evaluate the learned representations on NYUd (Silberman et al., 2012) for depth prediction.

yields no additional improvements. These findings extend the applicability of our conclusions from contrastive SSLs to a broader range of SSL methods under the instance consistency paradigm.

### B.2 Validation Across Diverse Tasks

To validate our findings to more downstream tasks, we evaluate the pre-trained models on COCO for object detection and instance segmentation and NYUd for depth prediction as mentioned in Appendix A.2, with MoCo-v2 framework pre-trained on COCO and ImageNet-100.

**Results.** As presented in Table D, results on COCO object detection and instance segmentation indicate that using smaller crop scales and zero overlapping continues to enhance baseline performance, with no added benefit from combining the two configs. Depth prediction results in Table E also align closely with the observations in classification and object detection tasks, exhibiting a clear U-curve as shown in Figure A due to the use of RMSE metric.

### B.3 Validation Across Diverse Experimental Settings

To validate our findings with diverse training conditions, we examine the pre-trained models with various experimental settings as follows:

**Full-Tuning $vs$ Frozen-Backbone Tuning.** Typically, downstream evaluations involve full fine-tuning of the backbone to adapt the pre-trained model for task-specific performance. To better isolate and preserve the learned representations from SSL pre-training, we evaluate models under a frozen-backbone setting, where only the task-specific head is fine-tuned while backbone weights remain fixed. This allows us to more

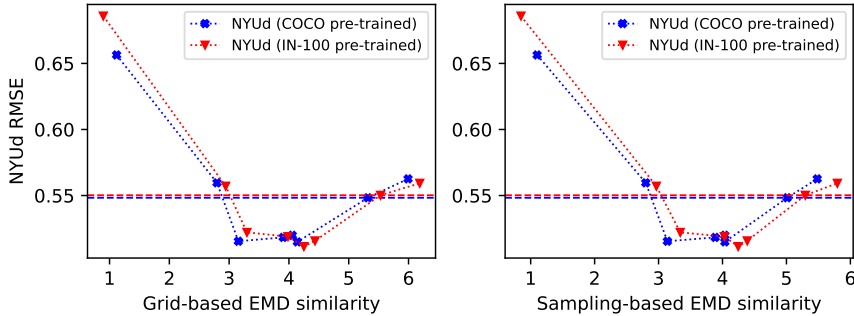

Figure A: **EMD similarity versus depth prediction results.** The similarity scores between views are plotted against depth prediction results. Note that RMSE is used for evaluation metric, thus results exhibit a clear U-curve.

| Config | COCO | | ImageNet-100 | |
|---|---|---|---|---|
| | VOC-0712 | DOTA-v1.0 | VOC-0712 | DOTA-v1.0 |
| random init. | 33.85 | 26.24 | 33.85 | 26.24 |
| Lower Bound | 18.63 $^{-40.65}$ | 14.63 $^{-26.57}$ | 19.65 $^{-40.71}$ | 15.42 $^{-27.12}$ |
| Baseline | 59.28 | 41.20 | 60.36 | 42.54 |
| Spatial Ovlp. $= 0$ | 62.35 $^{+3.07}$ | 43.85 $^{+2.65}$ | 62.39 $^{+2.03}$ | 44.04 $^{+1.50}$ |
| Only Bg | 63.16 $^{+3.88}$ | 44.05 $^{+2.85}$ | 63.41 $^{+3.05}$ | 44.18 $^{+1.64}$ |
| Smaller Crop | 62.68 $^{+3.40}$ | 42.39 $^{+1.19}$ | 63.16 $^{+2.80}$ | 43.91 $^{+1.37}$ |
| Smaller Crop$^{\dagger}$ | 60.25 $^{+0.97}$ | 41.30 $^{+0.10}$ | 60.96 $^{+0.60}$ | 42.89 $^{+0.35}$ |

Table F: **Object detection results under frozen-backbone tuning with MoCo-v2 (Chen et al., 2020b) pre-trained on COCO (Lin et al., 2014) and ImageNet-100 (Deng et al., 2009).**

directly observe the impact of instance consistency and diversity between positive pairs from pre-training. In this setup, as shown in Table F, we find similar trends to the full-tuning setting: using smaller crops and zero overlapping outperform the baseline, with no added gain from combining both. The performance gaps between baseline and other configurations are more pronounced in this setup, reinforcing that the observed performance improvements are due to different positive pair selection during SSL pre-training rather than downstream adaptation. This further supports our findings of the necessity of instance consistency and diversity between positive pairs for effective SSL.

**Transfer *vs* In-Distribution Evaluation.** In previous discussions, we primarily evaluate our findings using transfer learning tasks, where models are pre-trained on one dataset and fine-tuned on a different downstream one. To determine whether our insights hold for in-distribution tasks, where pre-training and evaluation on the same dataset, we conduct experiments where both pre-training and evaluation are performed on COCO (Lin et al., 2014). As stated in Appendix B.2, results in Table D indicate that using smaller crop scales and zero overlapping continues to enhance baseline performance, with no added benefit from combining the two configurations. This pattern mirrors the observation in transfer learning tasks and aligns with EMD measurement, reinforcing that our findings on instance consistency and view diversity are robust across both transfer and in-distribution scenarios.

**Short *vs* Long Epochs.** Initial experiments use a pre-training duration of 100 epochs, a relatively short period for SSL pre-training, which often performs longer duration to ensure effective training. To verify

| Config | 100 epochs | 200 epochs | 400 epochs |
|---|---|---|---|
| Baseline | 73.32 | 76.34 | 78.67 |
| Smaller Crop | 74.58 $^{+1.26}$ | 77.52 $^{+1.18}$ | 79.75 $^{+1.08}$ |
| Spatial Ovlp. $= 0$ | 74.90 $^{+1.58}$ | 77.90 $^{+1.56}$ | 79.99 $^{+1.32}$ |

Table G: **Object detection results with extended training epochs.**

| | COCO | | | | | | ImageNet-100 | | | |
| Config | VOC-0712 | | | DOTA-v1.0 | | | VOC-0712 | | DOTA-v1.0 | |
| | GT | MC | SS | GT | MC | SS | MC | SS | MC | SS |
|---|---|---|---|---|---|---|---|---|---|---|
| Baseline | 73.32 | | | 54.38 | | | 73.91 | | 55.30 | |
| Inst. *vs* Bg | 74.15 +0.83 | 74.12 +0.80 | 74.18 +0.86 | 55.47 +1.09 | 55.68 +1.30 | 55.56 +1.18 | 74.35 +0.44 | 74.30 +0.39 | 56.65 +1.35 | 55.94 +0.64 |
| Only Bg | 74.73 +1.41 | 74.52 +1.20 | 74.66 +1.34 | 55.34 +0.96 | 55.32 +0.94 | 55.41 +1.03 | 74.43 +0.52 | 74.44 +0.53 | 56.65 +1.35 | 56.37 +1.07 |

Table H: **Object detection results with different pseudo mask generation methods with MoCo-v2 (Chen et al., 2020b) pre-trained on COCO (Lin et al., 2014) and ImageNet-100 (Deng et al., 2009).** **MC** and **SS** refer to the pseudo mask generation methods MaskCut (Wang et al., 2023) and Selective Search (Uijlings et al., 2013), respectively.

| Crop Scale | VOC-0712 | DOTA-v1.0 |
|---|---|---|
| Baseline | 73.91 | 55.30 |
| $s = (0.18, 0.9)$ | **74.49** | **56.26** |
| $s = (0.16, 0.8)$ | 73.97 | 55.66 |
| $s = (0.14, 0.7)$ | 73.98 | 55.34 |
| $s = (0.12, 0.6)$ | 74.06 | 55.23 |
| $s = (0.1, 0.5)$ | 74.04 | 55.74 |
| $s = (0.08, 0.4)$ | 73.84 | 55.51 |

Table I: **Object detection results with varied crop scales with MoCo-v2 (Chen et al., 2020b) pre-trained on ImageNet-100 (Deng et al., 2009).** The Baseline config uses a scaling range of $s = (0.2, 1.0)$. Optimal performance is achieved with a scaling range of $s = (0.18, 0.9)$.

whether our findings hold with more epochs, we extend the pre-training duration to 200 and 400 epochs. As shown in Table G, with more training epochs, using smaller crop scales and enforcing zero overlapping consistently enhance performance over the baseline. This reinforces that our observations regarding instance consistency and view diversity remain consistent across different training durations.

## B.4 Additional Ablation Studies

We provide additional ablation studies regarding the implementation details for proposed configs as follows.

**Ablation Studies on Pseudo Masks.** We conduct ablation experiments comparing two pseudo mask generation methods, as shown in Table H. Rather than relying on explicit bounding box or mask accuracy, we evaluate pseudo-mask quality via their impact on downstream SSL performance. Specifically, the experiments are conducted with MoCo-v2 framework and evaluated on VOC-0712 and DOTA-v1.0 under three mask sources: Ground Truth (GT), MaskCut (MC), and Selective Search (SS).

As the results show, both pseudo-mask methods closely track the performance of ground-truth masks across diverse configurations and datasets. This supports our claim that the generated pseudo masks are sufficiently reliable to approximate the real instance locations for the purpose of our controlled analysis. We adopt **MC** in all our experiments.

**Ablation Studies on Crop Scales.** To optimize downstream task performance, we conduct the ablation experiments on the selection of crop scales, focusing specifically on the scaling range used in ***Smaller Crop*** configuration. The scale used in ***Larger Crop*** is fixed to $s = (0.4, 1.0)$, which doubles the default setting.

For COCO, we derive the scaling range directly from the average object instance size provided in the dataset annotations, resulting in a range of $s = (0.08, 0.4)$. For ImageNet-100, we vary the scaling range incrementally from 0.08 to 0.2, with a step size of 0.02, to systematically observe changes in downstream performance for object detection. The experiments are conducted with MoCo-v2 framework and evaluated on VOC-0712 and DOTA-v1.0. As shown in Table I, applying the scaling range of $s = (0.18, 0.9)$ achieves optimal results for both downstream tasks. Based on these results, we adopt this scaling range in all our experiments.

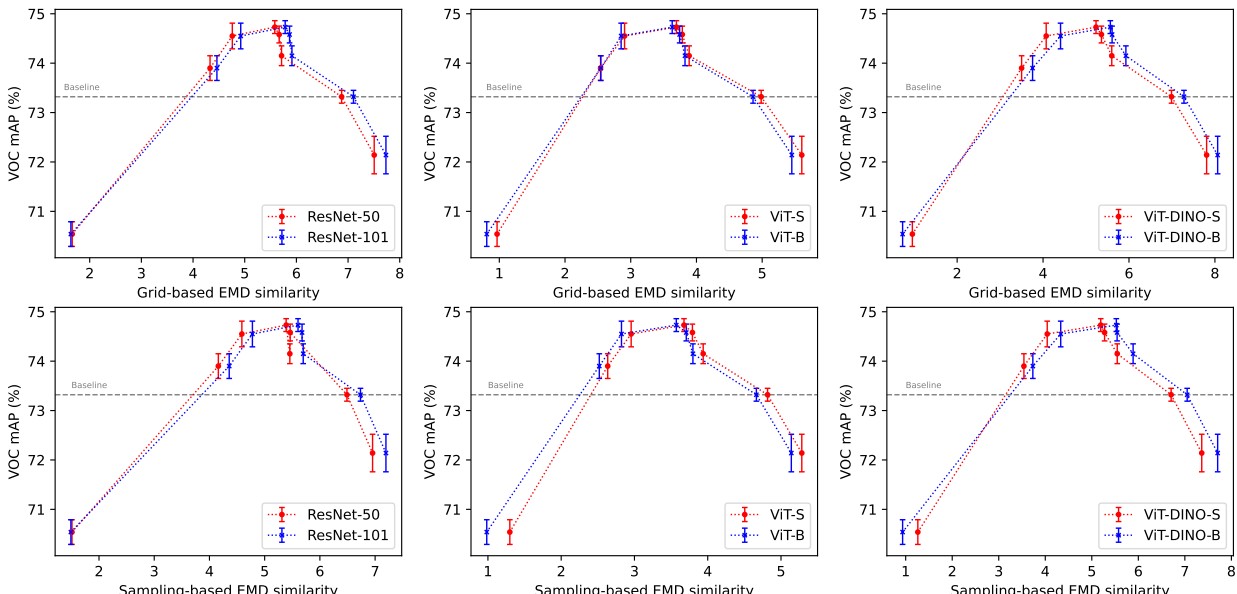

Figure B: **EMD plots with different feature encoders.** The similarity scores between views are plotted against object detection results. EMD scores are computed with six different feature encoders: ResNet-50, ResNet-101, ViT-S, ViT-B, ViT-DINO-S, and ViT-DINO-B. Baseline configuration is highlighted for reference.

### B.5   Additional Results on EMD-based Estimator

We provide additional validation results to support the reliability of the EMD-based similarity score as an estimator of view diversity.

We use pre-trained ResNet-50 (He et al., 2016) and ViT-S (Dosovitskiy et al., 2021) to extract features for computing the EMD-based score for MoCo-v2 and DINO, respectively. Figure C presents the relationships between EMD scores with object detection accuracy and classification accuracy for models pre-trained on ImageNet-100 (Deng et al., 2009). Across all tasks, our results consistently reveal a clear reverse-U curve under the two proposed cropping strategies, reinforcing our findings that EMD can serve as an effective estimator of view diversity across different data sources.

Furthermore, we conduct a cross-architecture validation experiment by computing EMD using six different feature encoders: ResNet-50, ResNet-101, ViT-S, ViT-B, ViT-DINO-S, and ViT-DINO-B. As illustrated in Figure B, all plots exhibit a clear and consistent reverse-U curve under the two proposed cropping strategies, indicating that a moderate level of view diversity, as estimated via EMD, correlates robustly with optimal downstream performance, regardless of the encoder used. This consistency across architectures further emphasizes the generalizability of our findings.

### B.6   Additional Results on Medical Imaging Domain

We provide additional validation results on medical imaging domain to further evaluate the generalizability of our findings. Specifically, we conduct the proposed controlled experiments on the NIH Chest X-ray dataset (Wang et al., 2017) with MoCo-v2. Following Sowrirajan et al. (2021), we explore two settings for SSL pre-training: (1) training from scratch, and (2) training with ImageNet-pretrained weights, which are known to provide stronger representations and initializations for downstream medical tasks.

We evaluate the performance of the pre-trained models with the disease classification task on the same dataset. To maintain consistency with standard classification evaluation, we filter out samples with multiple or missing labels, and report the Top-1 classification accuracy. As shown in Table J, the trends observed in the medical domain align with those from natural image datasets: Increasing diversity between positive pairs consistently

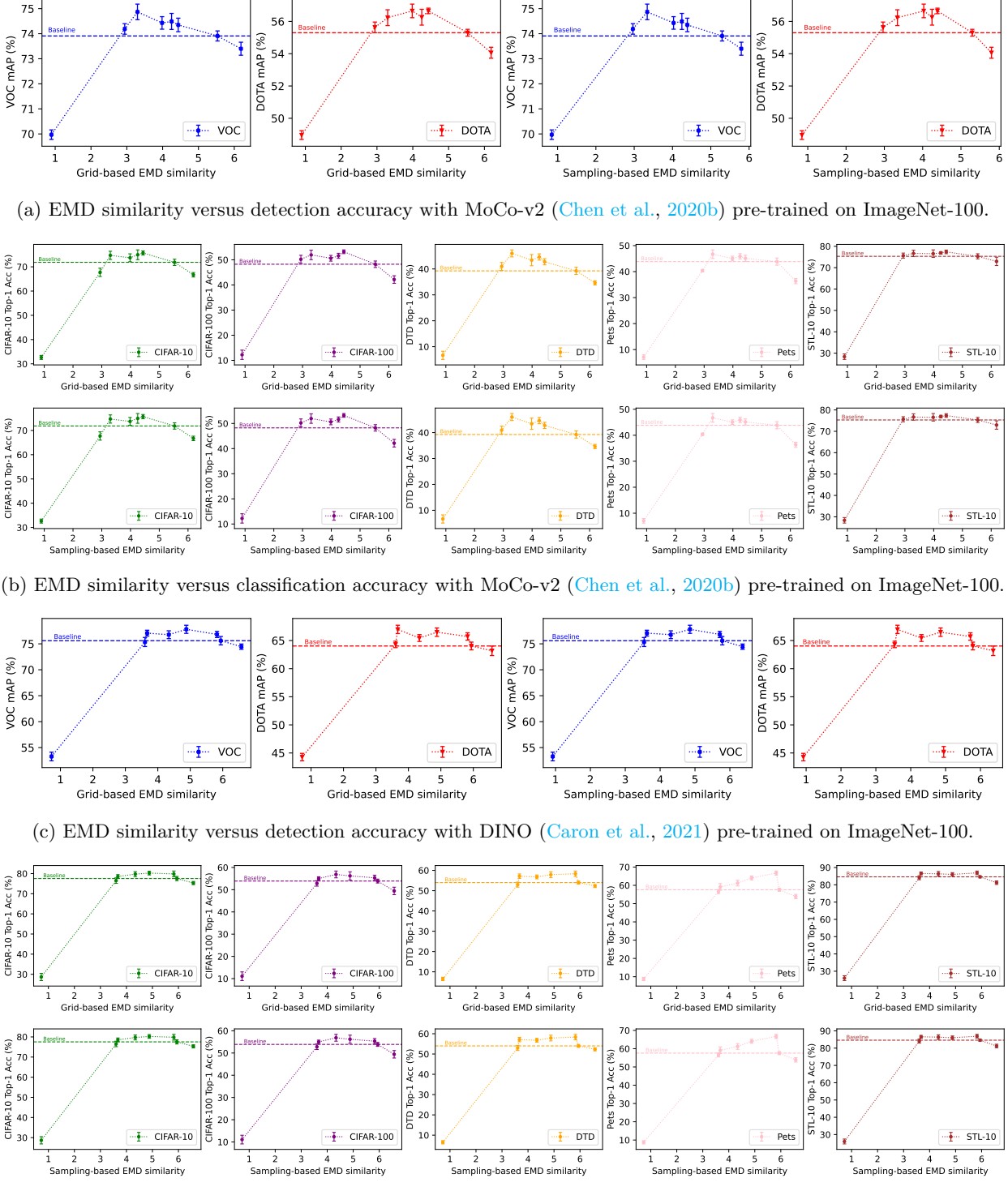

(a) EMD similarity versus detection accuracy with MoCo-v2 (Chen et al., 2020b) pre-trained on ImageNet-100.

(b) EMD similarity versus classification accuracy with MoCo-v2 (Chen et al., 2020b) pre-trained on ImageNet-100.

(c) EMD similarity versus detection accuracy with DINO (Caron et al., 2021) pre-trained on ImageNet-100.

(d) EMD similarity versus classification accuracy with DINO (Caron et al., 2021) pre-trained on ImageNet-100.

Figure C: **EMD similarity versus detection and classification accuracy.** The similarity scores between views are plotted against object detection and classification results. Baseline configuration is highlighted for reference.

| Config | Baseline | Lower Bound | Spatial Ovlp. $= 0$ | Inst. *vs* Bg | Only Bg | Larger Crop | Smaller Crop | Smaller Crop$^{\dagger}$ |
|---|---|---|---|---|---|---|---|---|
| w/o pre-train | 31.8 | 27.8 $_{-4.1}$ | 33.9 $_{+2.1}$ | 33.9 $_{+2.1}$ | 33.7 $_{+1.9}$ | 29.1 $_{-2.7}$ | 33.0 $_{+1.1}$ | 30.5 $_{-1.4}$ |
| with pre-train | 39.7 | 29.3 $_{-10.4}$ | 41.7 $_{+2.1}$ | 41.9 $_{+2.2}$ | 42.0 $_{+2.4}$ | 35.7 $_{-4.0}$ | 41.2 $_{+1.5}$ | 38.1 $_{-1.5}$ |

Table J: **Classification results with MoCo-v2 (Chen et al., 2020b) pre-trained on NIH Chest X-ray (Wang et al., 2017).** We freeze the pre-trained weights of the SSL backbone and train a supervised linear classifier to evaluate the learned representations on the same dataset following Sowrirajan et al. (2021).

improves downstream performance; excessive diversity, however, yields no additional improvements. These results reinforce the robustness of our findings and suggest their applicability extends beyond natural images to specialized domains such as medical imaging.

### B.7 Additional Results on Time Cost for EMD Computation

We provide additional results to quantify the computational cost of computing EMD and clarify its feasibility as an offline metric for guiding optimal positive pair selection in SSL pre-training.

**Robustness to Data Fraction.** To reduce the cost of full-dataset computation, we first examine the stability of EMD scores when being computed on only a small subset of the whole dataset. As shown in Table K, even with just 0.1% of the data, EMD scores remain consistent and are well-separated across different configs. This confirms that EMD can be reliably estimated with a minimal fraction of data, significantly reducing the computation burden.

| Data Fraction $\rightarrow$ | 100% | 50% | 20% | 10% | 1% | 0.1% |
|---|---|---|---|---|---|---|
| EMD (Baseline) | 0.6876 | 0.6878 | 0.6889 | 0.6880 | 0.6869 | 0.6872 |
| EMD (Spatial Ovlp. $= 0$) | 0.4329 | 0.4326 | 0.4330 | 0.4337 | 0.4328 | 0.4332 |

Table K: **Grid-based EMD similarity scores using ResNet-50 encoder pre-trained on COCO.**

**Wall-Clock Time Comparison.** We then compare the time required to compute EMD (with both full and 0.1% data) against the time required for standard SSL pre-training using MoCo-v2 on COCO as shown in Table L. Considering evaluating one SSL setting typically requires 100 epochs of pre-training, even computing EMD on the full dataset is already 10× faster, and using 0.1% data yields a speed-up of over 10,000×. This confirms that EMD introduces negligible overhead, especially when used as a one-time offline estimator.

| | EMD 100% data | EMD 0.1% data | SSL pre-training 1 epoch | SSL pre-training 100 epochs |
|---|---|---|---|---|
| Time (mins) $\downarrow$ | 10.17 | **0.01** | 1.07 | 106.67 |

Table L: **Wall-clock time cost comparison for EMD computation.** Time are measured with the batch size of 256 on 8 NVIDIA RTX A6000 GPUs and AMD EPYC 7543 32-Core CPU, with EMD solver from OpenCV.

These two observations confirm that EMD is both effective and efficient for quantifying view diversity, making it a practical tool for guiding positive pair selection in real-world SSL pipelines.

### B.8 Ablation on Impact of Memory Bank in MoCo-v2

We conduct additional ablation experiments to analyze the impact of memory bank in MoCo-v2. Specifically, the MoCo-v2 models are pre-trained on COCO across varying memory bank sizes, and evaluated on VOC-0712 for object detection. As shown in Figure D, models pre-trained on COCO generally fail to benefit from larger memory banks, in contrast to the trend observed on ImageNet, as reported in Figure 3 of He et al. (2020).

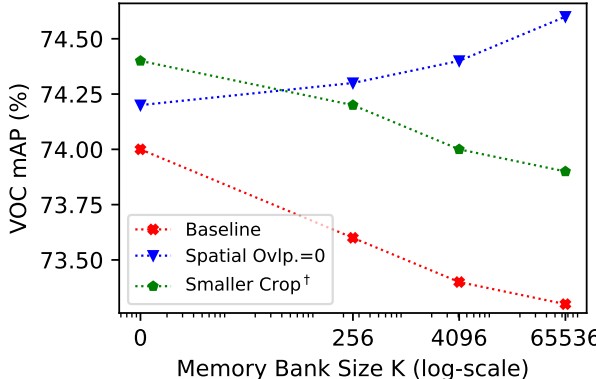

Figure D: **Effect of memory bank size with MoCo-v2 pre-trained on COCO.**

Notably, we observe that increasing view diversity within the optimal EMD range enhances the effectiveness of utilizing larger memory banks, while excessive diversity remains ineffective. This observation further demonstrates the broader applicability of our findings in understanding SSL mechanisms on diverse data sources.

## C   Further Discussions

### C.1   Gains Observed and Their Significance

Our experiments highlight consistent and meaningful performance improvements across different positive pair selection configurations in SSL. While modifying views in augmentation perspective naturally yields relatively modest gains, as reported in prior works Peng et al. (2022) and Van Gansbeke et al. (2021), our results demonstrate that targeted control of view diversity produces improvements with practical significance. Specifically, our proposed configurations yield:

- ∼1.5% mAP gain on VOC detection;

- over 3× higher gains on COCO detection than reported in Peng et al. (2022);

- and 3-5% accuracy improvement on classification tasks.

These improvements remain consistent even under longer training durations, as shown in Table G, reinforcing the robustness of our findings. The consistent performance gap across datasets and tasks demonstrates the importance of view diversity control in SSL, especially for non-iconic or complex data sources.

### C.2   Applicability to Reconstruction-based SSLs

Our study specifically focuses on investigating *the necessity of instance-level consistency in SSLs*, which serves as a core assumption in contrastive- and distillation-based SSLs, such as MoCo-v2 and DINO. These methods explicitly treat each image as a separate class and primarily rely on the consistency between augmented views of the same image to learn meaningful representations.

In contrast, reconstruction-based SSLs like MAE (He et al., 2022) and SimMIM (Xie et al., 2022b) are designed with a fundamentally different objective: to reconstruct the masked parts of an image, without requiring or benefiting from carefully designed view augmentations. As such, they do not rely on instance-level consistency and thus fall outside the scope of our study.

For this reason, we focus our validation on contrastive- and distillation-based methods, which are the most representative paradigms aligned with our research objective.

### C.3 Clarifying Scope and Relationship to Prior Work

We clarify the distinctions between our contributions and related prior works below.

**Comparison with Tian et al. (2020).** While both works observe a reverse-U-shaped relationship between view diversity and SSL downstream performance, our study addresses a fundamentally different research question and offers several novel contributions:

1) **Different Research Focus: Instance Consistency *vs*. View Informativeness.** InfoMin (Tian et al., 2020) investigates the informativeness of views and hypothesizes that optimal pair of views should share minimal but sufficient mutual information for learning useful representations. In contrast, our work focuses on investigating the *assumption of instance-level consistency*. This is particularly relevant for non-iconic data, where two augmented views from the same image may not guarantee to contain the same object or share consistent semantic information.

2) **Broader and More Realistic Evaluation Settings.** InfoMin evaluates view informativeness solely on iconic data, where the consistency between object instances in different crops is largely guaranteed. We go beyond this by performing controlled experiments on both iconic and non-iconic datasets, across multiple SSL frameworks. This broadens the practical relevance of our findings and systematically studies the effectiveness of SSL across real-world data sources.

3) **From Qualitative Observation to Quantitative Estimation via EMD.** InfoMin proposes a qualitative hypothesis about the existence of an optimal *sweet point* in view diversity, but does not offer a concrete metric to quantify it. We go a step further by introducing EMD as a quantitative and generalizable tool to estimate view diversity and identify this *sweet spot* reliably across diverse datasets, SSL methods, and downstream tasks.

4) **Temporal Relevance and Novel Insights.** InfoMin is grounded in earlier SSL settings based on iconic data and instance-consistent views. Our work offers new insights in the current context, where non-iconic, uncurated, and large-scale web datasets are increasingly used for SSL pre-training. In such settings, ensuring instance consistency is no longer practical, and we demonstrate that SSL can still remain effective, thereby providing an important and timely contribution to modern representation learning.

**Comparison with Moutakanni et al. (2024).** This work demonstrates that data augmentations may become less critical in large-data-scale SSL settings. While we agree with this observation, we emphasize that the focus of our work is fundamentally different. Specifically, our study investigates the *assumption of instance consistency* in SSLs. Specifically, we aim to understand how the degree of instance consistency affects the quality of the obtained representations, particularly when pre-training on non-iconic data where instance-level semantic alignment is not guaranteed.

Rather than studying the necessity of augmentation at scale, our work investigates how modern SSL frameworks generalize to broader and more realistic training data sources, including uncurated and diverse scenarios. In this sense, our analysis complements the discussion in Moutakanni et al. (2024) by targeting a different axis of the SSL landscape.

## D Future Work

Our study provides an initial investigation into the role of instance consistency and diversity between positive pairs in SSL. We highlight several directions for future research:

**Broader Evaluation Across SSL Methods.** While we validate our findings on contrastive (Chen et al., 2020b) and distillation-based (Caron et al., 2021) SSL methods, future work could examine whether similar trends hold for other paradigms such as SimCLR (Chen et al., 2020a), SwAV (Caron et al., 2020), BYOL (Grill et al., 2020), and NNCLR (Dwibedi et al., 2021). Understanding how view diversity influences learning across these different objectives could further generalize our insights.

**Scaling to Larger Pre-training Datasets.** Our experiments primarily use moderate-sized datasets such as COCO (Lin et al., 2014) and ImageNet-100 (Deng et al., 2009). Investigating whether the observed consistency-diversity trade-off holds on larger and more diverse datasets like OpenImages (Kuznetsova et al., 2020) would validate scalability and offer insights into SSL behavior under more realistic data regimes.

**Toward Theoretical Insights.** While our findings are grounded in extensive empirical analysis, a theoretical understanding of how mutual information, instance consistency, and view diversity interact in SSL remains an open question. Developing such a theoretical framework could deepen our understanding of view-based supervision and guide the design of more adaptive or learnable augmentation strategies.

