# OpenReview forum: "Beyond Instance Consistency: Investigating View Diversity in Self-supervised Learning"
_TMLR — Accepted by TMLR_

### Review · Reviewer_oinq · 2025-06-24

**Summary Of Contributions:**

This work conducts a comprehensive experimental evaluation on the creation of positive views in visual SSL, precisely: 1) Instance Diversity of views is controlled using the bounding boxes (ground truth or based on prediction from pretrained neural net object detectors), 2) Scale Diversity consistns of a group of exprimental that explicitly controlled the cropping scale range in the randomresize crop transformation. Finally, using a supervised, pre-trained ResNet50, the authors attempt to quantify a measure of mutual information based on the way the views are generated.  Similar to [7], a u-shaped behaviour is observed, meaning that view diversity can be enhanced to some extent while leading to superior downstream performance (object detection + classification) compared to the current augmentation pipeline, but not to a large extent.

**Audience:**

Yes

**Claims And Evidence:**

Yes

**Requested Changes:**

Apart from my main concerns and specifically d,e,f:
- Nearest-Neighbor Contrastive Learning would have been a valuable addition to the study, especially in conjunction with the MOCO baseline, which utilizes a memory bank.

- In general, there are a lot of things to improve in the manuscript, for instance, sec 3 is trivially explained in my humble opinion and can be replaced by more experiments (as I wrote in minor concerns)

- To repeat my main concerns concerning EMD, the presentation of EMD as an alternative to the infonce mutual info estimation in [7] must be clearly explained. The differences in the papers, scopes and investigation should be crystal clear in the next version of the manuscript. On top the EMD section must be more meticulously written and the reliance to a supervised models must be tested with an label-free pretrained model. A pseudocode for EMD would probably be more useful compared to the algorithm in its current form where many details are missing for other researchers to plug their pipeline and test it.

**Strengths And Weaknesses:**

## Strengths
- The evaluations are diverse in many crucial aspects: pretraining dataset, SSL objective/pipeline, architecture, and downstream dataset tasks.

- The paper is overall well-written, and the message is clearly conveyed to the reader.

## MAJOR concerns

a) **Fine-grained classification datasets**: The reported results on the classification datasets do not correspond to the standard fine-grained classification benchmarks, such as CUB, Car, Dog, Flower, see citations below. The textures dataset (DTD) is certainly not in the category of fine-grained classification. refs [1-4].


b) **Data determines the scope of the analysis**. While I see some benefit from smaller-scale analysis on the sensitivity of the chosen augmentations, this analysis is heavily dependent on the dataset (number of samples, quality of data, number of semantic categories). The authors must state this as a limitation/scope of their experimental analysis. In [6], it is recently shown that the augmentations are due to the lack of data, and when scaling from Imagenet1k to imagenet21k (>10M samples), augmentations become less relevant. I do still agree on the presented way of separating iconic vs non-iconic datasets.


c) Crop scale ranges in the Scale Diversity category: I would at least expect the authors to share the final values of the crop scales used in Section 4.2, where they refer to smaller and larger crops. This must be in the main text.

d) EMD. There is not enough detailed description in the main manuscript for the EMD calculations, which is supposed to be a major contribution of this work. For me, it’s not quite clean how this computation is done. How many samples are used? How many random augmentations per image? The algorithm only shows what’s happening once N features of dimension D are available from 2 views.


e) Connections, differences, and overlap with proximal work [7]. This work does not sufficiently describe the overlap with the existing work. How are the estimations of mutual information related compared to [7]? It is not enough to write our results align with [7]. If both manuscripts explore a similar topic, why conduct similar experiments and end up in similar findings 4 years later? This is a major concern for me. The u-shaped behaviour in Figure 4 has also been shown in [7].

f) Limited predictive power due to supervised ResNet-50 for EMD. The proposed EMD metric uses a supervised metric, which cancels out the idea of estimating the mutual info between views for self-supervised learning. An off-the-shelf feature extractor, such as dinov2 trained on large-scale and diverse unlabelled data, would make more sense here, in my view.

h) Section/paragraph: “Diverse Tasks & Experimental Settings.” In which experiments is this paragraph referring to? Are the authors discussing results that are related to the main text or supplementary? Cross-referencing is missing here.


### MINOR concerns

- Table 2: Which metric is used for object detection?

- As far as I understand, the supplementary material should have been submitted as a separate file and not at the end of the main manuscript.

- Results on ImageNet-1k in the supplementary. “ Results for SSL pre-trained on ImageNet (Deng et al., 2009) are provided in the supplementary materials.” Where are these results precisely?

The description of MOCO and DINO is trivial. Better to remove this part or write a self-sufficient preliminary explanation in the supplementary.

## References

[1] Wah, C., Branson, S., Welinder, P., Perona, P., & Belongie, S. (2011). The caltech-ucsd birds-200-2011 dataset.

[2] Krause, J., Stark, M., Deng, J., & Fei-Fei, L. (2013). 3d object representations for fine-grained categorization. In Proceedings of the IEEE international conference on computer vision workshops (pp. 554-561).

[3] Dataset, E. (2011). Novel datasets for fine-grained image categorization. In First workshop on fine grained visual categorization, CVPR. Citeseer. Citeseer. Citeseer (Vol. 5, No. 1, p. 2).

[4] Nilsback, M. E., & Zisserman, A. (2008, December). Automated flower classification over a large number of classes. In 2008 Sixth Indian conference on computer vision, graphics & image processing (pp. 722-729). IEEE.

[5] Dwibedi, D., Aytar, Y., Tompson, J., Sermanet, P., & Zisserman, A. (2021). With a little help from my friends: Nearest-neighbor contrastive learning of visual representations. In Proceedings of the IEEE/CVF international conference on computer vision (pp. 9588-9597).

[6] Moutakanni, T., Oquab, M., Szafraniec, M., Vakalopoulou, M., & Bojanowski, P. (2024, December). You Don't Need Domain-Specific Data Augmentations When Scaling Self-Supervised Learning. In NeurIPS 2024-The Thirty-Eighth Annual Conference on Neural Information Processing Systems.

[7] Tian, Y., Sun, C., Poole, B., Krishnan, D., Schmid, C., & Isola, P. (2020). What makes for good views for contrastive learning?. Advances in neural information processing systems, 33, 6827-6839.

---

> ### Author Response · Authors · 2025-08-15
> **Response to Reviewer oinq (1/3)**
>
> We thank the reviewer for the constructive feedback and the insightful questions. We would like to address your concerns below:
>
> > [W1] Clarifying the Classification Dataset Scope.
>
> We thank the reviewer for pointing this out.
>
> To avoid potential confusion, we have updated the term "fine-grained classification" to "classification" in the **revised version**.
>
> > [W2] Clarifying the Scope and Relationship to Prior Work [1].
>
> We thank the reviewer for highlighting [1], which demonstrates that augmentations may become less critical in large-data-scale SSL settings.
> While we agree with this observation, we emphasize that the focus of our work is fundamentally different.
>
> Specifically, our study investigates the assumption of *instance consistency* in SSLs.
> Specifically, we aim to understand how the degree of instance consistency affects the quality of the obtained representations, particularly when pre-training on non-iconic data where instance-level semantic alignment is not guaranteed.
>
> **Rather than studying the necessity of augmentation at scale, our work investigates how modern SSL frameworks generalize to broader and more realistic training data sources, including uncurated and diverse scenarios.**
> In this sense, our analysis complements [1] by targeting a different axis of the SSL landscape.
>
> We appreciate the reviewer’s thoughtful feedback and have included the above discussion in **Appendix Section C.3** in the **revised version**.
> Additionally, we acknowledge the value of future studies to extend our current analysis to larger-scale data, and we plan to explore this in follow-up work.
>
> > [W3] Clarification on the Crop Scale Settings.
>
> We thank the reviewer for pointing this out.
> Specifically, we carefully adjust the crop scales with different pre-training datasets as mentioned in **Appendix Section A.1**.
> - For COCO, we use the average object instance size derived from dataset annotations, resulting in:
>     - Smaller crop scale: (0.08, 0.4)
>     - Larger crop scale: (0.4, 1.0)
> - For ImageNet, we adopt:
>     - Smaller crop scale: (0.18, 0.9)
>     - Larger crop scale: (0.4, 1.0)
>
> We provide detailed ablation studies examining the impact of crop scales in **Appendix Section B.4**, which further supports our design choices.
>
> > [W4] Description of EMD Calculation and Computational Cost.
>
> We thank the reviewer for the feedback and provide further clarification below.
>
> We include the detailed algorithm for EMD calculation in the **Appendix Section A.3**.
> Specifically, EMD is computed between two augmented views of the *same image* when conducting SSL pre-training.
> We use the proposed *Grid-based* or *Sampling-based* strategies to extract patch-level features using a frozen encoder.
> EMD is then calculated over the obtained patches.
>
> To evaluate the computational cost, we assess how much data is needed to provide consistent EMD estimates.
> The results below use *grid-based* EMD with ResNet-50 encoder pre-trained on COCO:
>
> |Data Fraction →|100%|50%|20%|10%|1%|0.1%|
> |-|-|-|-|-|-|-|
> |EMD (Baseline)|0.6876|0.6878|0.6889|0.6880|0.6869|0.6872|
> |EMD (Spatial Ovlp.=0)|0.4329|0.4326|0.4330|0.4337|0.4328|0.4332|
>
> As shown, even with just **0.1% of the data**, EMD scores remain consistent and are well-separated across configurations.
> This confirms that EMD can be **reliably estimated from a small subset of the training data**, substantially reducing the computation burden.
>
> We will include this discussion and the quantitative results in the **final version**.

---

> ### Author Response · Authors · 2025-08-15
> **Response to Reviewer oinq (2/3)**
>
> > [W5] Clarifying the Scope and Contributions Relative to InfoMin [2].
>
> We thank the reviewer for raising this important point regarding the relationship between our work and InfoMin [2].
>
> While both works observe a reverse-U-shaped relationship between view diversity and SSL downstream performance, our study addresses a fundamentally different research question and offers several novel contributions beyond [2]:
>
> - **Different Research Focus: Instance Consistency vs. View Informativeness**
>
>     InfoMin investigates the informativeness of views and hypothesizes that optimal pair of views should share minimal but sufficient mutual information for learning useful representations.
>
>     In contrast, our work focuses on investigating the *assumption of instance-level consistency*.
>     This is particularly relevant for non-iconic data, where two augmented views from the same image may not guarantee to contain the same object or share consistent semantic information.
>
> - **Broader and More Realistic Evaluation Settings**
>
>     InfoMin evaluates view informativeness solely on iconic data, where the consistency between object instances in different crops is largely guaranteed.
>
>     We go beyond this by performing controlled experiments on both iconic and non-iconic datasets, across multiple SSL frameworks.
>     This broadens the practical relevance of our findings and systematically studies the effectiveness of SSL across real-world data sources.
>
> - **From Qualitative Observation to Quantitative Estimation via EMD**
>
>     InfoMin proposes a qualitative hypothesis about the existence of an optimal "sweet point" in view diversity, but does not offer a concrete metric to quantify it.
>
>     We go a step further by introducing EMD as a quantitative and generalizable tool to estimate view diversity and identify this "sweet spot" reliably across diverse datasets, SSL methods, and downstream tasks.
>
> - **Temporal Relevance and Novel Insights**
>
>     InfoMin is grounded in earlier SSL settings based on iconic data and instance-consistent views.
>
>     Our work offers new insights in the current context, where non-iconic, uncurated, and large-scale web datasets are increasingly used for SSL pre-training.
>     In such settings, ensuring instance consistency is no longer practical, and we demonstrate that SSL can still remain effective, thereby providing an important and timely contribution to modern representation learning.
>
> We have explicitly included the above clarifications in in **Appendix Section C.3** in the **revised version** to better distinguish our contributions from prior work.
>
> > [W6] More Evaluations on EMD Computation.
>
> We appreciate the reviewer’s suggestion to further validate the robustness of EMD computation across different feature encoders.
>
> To address this, we provide a **cross-architecture validation** experiment where EMD is computed using various source feature encoders, and the results are correlated with the downstream performance of ResNet-based MoCo-v2 models.
>
> Specifically, we use the following **six feature encoders**: ResNet-50, ResNet-101, ViT-S, ViT-B, ViT-DINO-S, and ViT-DINO-B.
>
> As shown in **Appendix Figure C**, all six plots exhibit a clear and consistent reverse-U shape relationship, indicating that a moderate level of view diversity (as estimated via EMD) correlates with optimal downstream performance, regardless of the encoder used.
>
> This consistency across architectures further emphasizes the **generalizability** of our findings.
> We have included the above discussion in **Appendix Section B.5** in the **revised version**.

---

> ### Author Response · Authors · 2025-08-15
> **Response to Reviewer oinq (3/3)**
>
> > [W7 W10] Clarifying Diverse Settings and ImageNet Results.
>
> We thank the reviewer for pointing out the need for clearer cross-reference between the main text and supplementary materials.
>
> To clarify:
>
> - The discussion in **Section 4.4** refers to the validation experiments of our findings across diverse tasks and experimental settings, which are provided in **Appendix Section B.2 & B.3**.
> - Regarding the ImageNet results mentioned in Section 4.3, we specifically refer to the U-shape plots presented in **Figure B in Appendix Section B.5**.
>
> We have explicitly added these cross-references in the **revised version** to avoid any confusion.
>
> > [W8] Clarification on Experimental Setups.
>
> We appreciate the reviewer’s suggestion and will incorporate the requested clarifications in the **final version**.
>
> Specifically, for detection tasks, we report the mean Average Precision (mAP) as the evaluation metric.
>
> > [W9] Positioning of Supplementary Materials
>
> We will check with the Action Editor to confirm the appropriate formatting requirements for TMLR.
>
> > [C1] Further Validation Experiments with SSL w/o Memory Bank.
>
> We thank the reviewer for recommending the Nearest-Neighbor Contrastive Learning [3] as a valuable addition to our study.
> We agree that NN-CL presents an important baseline that could offer complementary insights.
>
> While we leave a full-scale exploration of NN-CL for future work, particularly due to the effort for conducting a comprehensive controlled studies.
> To partially address the reviewer’s concern regarding memory mechanisms in contrastive learning, we provide additional experiments by **removing or reducing the memory bank size** in our MoCo-v2 baseline:
>
> |memory bank size →|65536|4096|256|0 (No Memory)|
> |-|-|-|-|-|
> |Baseline|73.3|73.4|73.6|74.0|
> |Spatial Ovlp.=0|74.6|74.4|74.3|74.2|
> |Smaller Crop $^†$|73.9|74.0|74.2|74.4|
>
> Models are pre-trained on COCO using MoCo-v2 across varying memory bank sizes, and evaluated on VOC detection.
>
> Results indicate that
> - **Larger memory banks do not provide substantial gains on COCO**, in contrast to the trend observed in ImageNet in Figure 3 in [4].
> - Increasing view diversity within the optimal EMD range **enhances the effectiveness of larger memory banks**, while **excessive diversity remains ineffective**.
>
> We believe this observation further demonstrates the broader applicability of our findings in understanding SSL mechanisms on diverse data sources.
>
> We have included the above discussion in **Appendix Section B.6** in the **revised version** with a performance plot for better clarity.
>
> ---
> We thank the reviewer again for the valuable feedback and we hope our response can address your questions.
> If you have any further questions or concerns, we are very happy to answer.
>
> [1] Moutakanni, Théo, et al. You don’t need domain-specific data augmentations when scaling self-supervised learning.
>
> [2] Tian, Yonglong, et al. What makes for good views for contrastive learning?.
>
> [3] Dwibedi, Debidatta, et al. With a little help from my friends: Nearest-neighbor contrastive learning of visual representations.
>
> [4] He, Kaiming, et al. Momentum contrast for unsupervised visual representation learning.

---

### Review · Reviewer_vJZ6 · 2025-07-07

**Summary Of Contributions:**

The paper questions the instance-consistency assumption that dominates modern image-based self-supervised learning. Instead of insisting that two augmented views must depict the same object, the authors provide a systematic analysis of how much shared content is actually needed to learn useful representations. Their key contributions are:

1.  Using COCO and ImageNet-100, the paper shows that positive pairs containing only background pixels or even coming from disjoint image regions, can match or outperform the standard random-crop baseline on classification, detection, segmentation and depth tasks.

2. As the visual overlap between the two views decreases, accuracy first improves and then collapses, revealing an optimal moderate diversity regime rather than the extremes of identical or totally unrelated crops.

3.  The paper quantifies diversity with the 1-Wasserstein distance
$W_1\bigl(P_{\text{view}\,1},\,P_{\text{view}\,2}\bigr)$,
and shows that values around 0.3–0.4 consistently coincide with the performance peak across 10 crop configurations, two SSL frameworks (MoCo-v2, DINO) and five downstream datasets.

4. The phenomena hold for both convolutional (ResNet-50) and transformer (ViT-S/16) backbones, and transfer from image-level classification to dense prediction tasks without retraining the backbone.

5. The authors propose a simple, task-agnostic crop-policy-tuning recipe: measure $W_1$ offline for candidate augmentations, select those in the mid-range band, and skip costly manual curation of object-centric images.

This indicates a possible relaxation of a central design constraint in SSL and suggest a principled knob for controlling view diversity, and demonstrate that robust representations can emerge even when positive pairs do not share a clearly delineated object.

**Audience:**

Yes

**Broader Impact Concerns:**

By relaxing the instance-consistency assumption, practitioners might transfer the method to medical, autonomous-driving or security domains without verifying that the learned features remain reliable when foreground-background statistics differ. Mis-calibration here could yield brittle models and harmful decisions. The paper should explicitly warn against blind deployment and suggest domain-specific validation.

Training on unfiltered Internet images can encode societal or demographic biases (e.g. under-representation of certain ethnicities). Because the method reduces the need for manual curation, the risk of latent bias propagation may rise. A bias audit protocol or mitigation advice would strengthen the Broader-Impact section.

**Claims And Evidence:**

Yes

**Requested Changes:**

1. Could you include an experiment on a medical imaging dataset such as NIH Chest X-ray 14 or CheXpert, using the provided lung or pathology bounding boxes.
It would be interest to tests whether the reverse, U diversity curve and the EMD "sweet-spot" generalise when foreground boundaries are weak and the global appearance is highly uniform.

2. For the principal results: Baseline, Peak, and Extreme-diversity points in Fig. 4, could you run each configuration at least three times and report mean $\pm$ standard deviation; add paired $t$-tests or bootstrapped confidence intervals where appropriate.

3. Could you provide wall-clock time and GPU/CPU specs for computing Wasserstein distances on COCO-train2017 with your 9-patch sampler, and compare it to one epoch of SSL pre-training.

4. Please state how many images rely on ground-truth boxes versus pseudo-boxes, describe the pseudo-box generator’s mAP, and discuss whether those boxes are strictly necessary for reproducing the main findings.

5. Release code and pretrained weights. Publishing the crop samplers, EMD scripts and checkpoint zoo would markedly ease adoption and further validation.

6. Could you re-compute the EMD–accuracy correlation with a larger encoder (e.g.\ ResNet-101 or ViT-B/16) or with different feature-grid resolutions to check whether the 0.3-0.4 optimum is scale-agnostic.

7.  Could you briefly speculate, supported by at least one pilot result, on whether masked-prediction or reconstruction-based SSL benefits from the same diversity regime, or explain why it might differ.

**Strengths And Weaknesses:**

Strengths

- The paper shows that self-supervised models do not need two crops of the same object to learn strong features, overturning a decade-old design heuristic.

- Five crop policies, two datasets (COCO, ImageNet-100), two SSL paradigms (MoCo-v2, DINO), two backbones (ResNet-50, ViT-S/16) and four downstream tasks (classification, detection, segmentation, depth) make the conclusions relativly strong.

- EMD is introduced as a simple quantitative proxy for "just-right" view diversity, letting users measure rather than guess their augmentations.

- The paper is well structured; figures intuitively illustrate the crop regimes and the reverse-U effect. Code snippets for the EMD calculation are promised, aiding reproducibility.

-  Relaxing the instance-consistency requirement means large, uncurated photo collections or even surveillance video frames, can be used without costly object-centric filtering.

Weaknesses

- Several diversity setups rely on ground-truth or pseudo bounding boxes, adding extra annotations or engineering to an ostensibly unsupervised study.

- All data are natural images; the findings are untested on more homogeneous or safety-critical domains (e.g., chest X-rays) where foreground boundaries are diffuse and background statistics differ markedly.

- Key curves are reported with single runs; confidence intervals or significance tests are missing, making it hard to gauge robustness.

- The manuscript omits wall-clock cost for computing EMD over large datasets, leaving open whether the "offline tuning" step is negligible or burdensome.

- The paper does not cover reconstruction-based, masked-prediction or multimodal SSL frameworks; it is unclear whether the same diversity sweet spot applies there.

- The Wasserstein distance is measured with a frozen ResNet-50; correlations might change with backbone scale or patch granularity, an aspect not explored.

- Source code does not seem to be provided

---

> ### Author Response · Authors · 2025-08-15
> **Response to Reviewer vJZ6 (1/3)**
>
> We thank the reviewer for the constructive feedback and the insightful questions. We would like to address your concerns below:
>
> > [W1 C4] Use of Ground-Truth Labels in Designed Configurations.
>
> We thank the reviewer for raising this important point.
>
> Our primary goal is to investigate *"the necessity of instance-level consistency in SSLs"* under controlled experiments.
> To this end, the use of ground-truth-based view generation is **solely intended for controlled experimental setups**, allowing us to systematically assess how varying levels of instance consistency between positive pairs affect SSL performance.
> These configurations are **not designed to serve as data augmentation pipelines** for practical use in SSL pre-training.
>
> To further assess the robustness of our findings, we have conducted ablation experiments on pseudo mask generators for ImageNet, using two popular methods: MaskCut [1] and Selective Search [2].
> These results are presented in **Appendix Section B.4**, with further discussion regarding the implementation details included in **Appendix Section A.1**.
>
> Moreover, rather than relying on explicit bounding box or mask accuracy, we evaluate pseudo-mask quality via their impact on downstream SSL performance.
> Below, we show object detection results using MoCo-v2 pre-trained on COCO under three mask sources: ground truth (GT), MaskCut, and Selective Search.
>
> |VOC-0712|GT|MaskCut [1]|Selective Search [2]|
> |-|-|-|-|
> |Baseline|73.3|-|-|
> |Inst. vs Bg|74.2 (+0.9)|74.1 (+0.8)|74.2 (+0.9)|
> |Only Bg|74.7 (+1.4)|74.5 (+1.2)|74.6 (+1.3)|
>
> |DOTA-v1.0|GT|MaskCut [1]|Selective Search [2]|
> |-|-|-|-|
> |Baseline|54.4|-|-|
> |Inst. vs Bg|55.5 (+1.1)|55.7 (+1.3)|55.6 (+1.2)|
> |Only Bg|55.3 (+0.9)|55.3 (+0.9)|55.4 (+1.0)|
>
> As the results show, both pseudo-mask methods closely track the performance of ground-truth masks under diverse configurations and datasets.
> This supports our claim that the generated pseudo masks are sufficiently reliable to approximate instance locations for the purpose of our controlled analysis.
>
> We will add the above discussion and the corresponding experiments in the **final version**.
>
> > [W2 C1] Validation on Medical Imageing Dataset.
>
> We thank the reviewer for the suggestion to validate our findings on medical imaging data.
>
> To evaluate the generalizability of our findings, we conducted controlled experiments on the NIH Chest X-ray dataset with MoCo-v2.
> Following [3], we considered two settings for pre-training: (1) training from scratch, and (2) using ImageNet-pretrained weights, which are known to provide stronger representations and initialization for downstream medical tasks.
>
> We evaluated the performance of the pre-trained models with the disease classification task on the same dataset.
> To maintain consistency with standard classification evaluation, we filter out samples with multiple labels or missing labels, and report the Top-1 classification accuracy.
>
> ||Baseline|Lower Bound|Spatial Ovlp.=0|Inst. vs Bg|Only Bg|Larger Crop|Smaller Crop|Smaller Crop $^†$|
> |-|-|-|-|-|-|-|-|-|
> |w/o ImageNet pretrained|31.8|27.8 (-4.1)|33.9 (+2.1)|33.9 (+2.1)|33.7 (+1.9)|29.1 (-2.7)|33.0 (+1.1)|30.5 (-1.4)|
> |with ImageNet pretrained|39.7|29.3 (-10.4)|41.7 (+2.1)|41.9 (+2.2)|42.0 (+2.4)|35.7 (-4.0)|41.2 (+1.5)|38.1 (-1.5)|
>
> As shown above, the trends observed in the medical domain align with those from natural image datasets:
> - **Increasing diversity** between positive pairs consistently enhances baseline performance.
> - **Excessive diversity**, however, yields no additional improvements.
>
> These results reinforce the robustness of our findings and suggest their applicability extends beyond natural images to specialized domains such as medical imaging.
>
> We will include these experiments and discussions in the **final version**.

---

> ### Author Response · Authors · 2025-08-15
> **Response to Reviewer vJZ6 (2/3)**
>
> > [W3 C2] Statistical Significance of the Reported Performance Improvements.
>
> We appreciate the reviewer’s concern regarding the statistical significance of the reported gains.
>
> All experiments in our study are repeated **three times**, and we report the average accuracy across these runs.
> Due to space constraints, we did not include the standard deviation values in the original submission.
>
> We have updated **standard deviations** for our main results (Figures 4 and B) in the **revised version**.
> These updated plots reflect the statistical variability and support the consistency of our findings across multiple runs.
>
> > [W4 C3] Time Comparison for EMD Computation.
>
> We thank the reviewer for pointing out the need to quantify the cost of computing EMD and clarify its practicality as an offline tuning step.
>
> (1) **Robustness to Data Fraction**: To reduce the cost of full-dataset computation, we first examine the stability of EMD scores when computed on only a subset of the whole dataset.
> The results below use *grid-based* EMD with ResNet-50 encoder pre-trained on COCO:
>
> |Data Fraction →|100%|50%|20%|10%|1%|0.1%|
> |-|-|-|-|-|-|-|
> |EMD (Baseline)|0.6876|0.6878|0.6889|0.6880|0.6869|0.6872|
> |EMD (Spatial Ovlp.=0)|0.4329|0.4326|0.4330|0.4337|0.4328|0.4332|
>
> As shown, even with just **0.1% of the data**, EMD scores remain consistent and are well-separated across configurations.
> This confirms that EMD can be **reliably estimated with minimal subset sampling**, substantially reducing the computation burden.
>
> (2) **Wall-Clock Time Comparison**: We compare the time required to compute EMD (with full and 0.1% data) against SSL pre-training using MoCo-v2 on COCO:
>
> ||EMD (100% data)|EMD (0.1% data)|SSL pre-training (1 epoch)|SSL pre-training (100 epoch)
> |-|-|-|-|-|
> |Time (mins) ↓|10.17|**0.01**|1.07|106.67|
>
> Time are measured with the batch size of 256 on 8 NVIDIA RTX A6000 GPUs and AMD EPYC 7543 32-Core CPU, with EMD solver from OpenCV.
>
> Considering evaluating one SSL setting typically requires 100 epochs of pre-training, even computing EMD on the full dataset is **~10× cheaper**, and using 0.1% data is **>10,000× cheaper**.
> This confirms that **EMD introduces negligible overhead**, especially when used as a one-time offline estimator.
>
> **Conclusion**: These findings confirm that EMD is both effective and efficient for quantifying view diversity, making it a practical tool for guiding positive pair selection in real-world SSL pipelines.
>
> We will include this discussion and the quantitative results in the **final version**.
>
> > [W5 C7] Applicability to Reconstruction-based SSLs.
>
> We appreciate the reviewer’s suggestion regarding the applicability of our findings to reconstruction-based SSLs.
>
> As clarified in **Section 2**, the focus of our work is to investigate *"the necessity of instance-level consistency in SSLs"*.
> This assumption is a central design principle in **contrastive- and distillation-based SSLs**, such as MoCo-v2 and DINO, which explicitly treat each image as a separate class.
> These methods primarily rely on consistency between augmented views to learn meaningful representations.
>
> In contrast, reconstruction-based SSLs like MAE [4] and SimMIM [5] are designed with a fundamentally different objective: to reconstruct the masked parts of an image, without requiring or benefiting from carefully designed view augmentations.
> As such, they do not rely on instance-level consistency and thus fall outside the scope of our study.
>
> For this reason, we focus our validation on contrastive- and distillation-based methods, which are the most representative paradigms aligned with our research objective.
>
> We hope the above clarification can address the reviewer's concern, and we have added the above discussion in **Appendix Section C.2** in the **revised version**.

---

> ### Author Response · Authors · 2025-08-15
> **Response to Reviewer vJZ6 (3/3)**
>
> > [W6 C6] More Evaluations on EMD Computation.
>
> We appreciate the reviewer’s suggestion to further validate the robustness of EMD computation across different feature encoders.
>
> To address this, we provide a **cross-architecture validation** experiment where EMD is computed using various source feature encoders, and the results are correlated with the downstream performance of ResNet-based MoCo-v2 models.
>
> Specifically, we use the following **six feature encoders**: ResNet-50, ResNet-101, ViT-S, ViT-B, ViT-DINO-S, and ViT-DINO-B.
>
> As shown in **Appendix Figure C**, all six plots exhibit a clear and consistent reverse-U shape relationship, indicating that a moderate level of view diversity (as estimated via EMD) correlates with optimal downstream performance, regardless of the encoder used.
>
> This consistency across architectures further emphasizes the **generalizability** of our findings.
> We have included the above discussion in **Appendix Section B.5** in the **revised version**.
>
> > [W7 C5] Release of Source Code and Pre-trained Models.
>
> We appreciate the reviewer’s suggestion and recognize the importance of supporting reproducibility.
>
> We will **try our best to release the implementation of the designed experiments**.
>
> > [Broader Impact Concerns] Expanding the Broader Impact Section
>
> We thank the reviewer for raising these important points regarding the broader impact of our work.
>
> We will explicitly address the mentioned concerns in the **final version**.
>
> ---
> We thank the reviewer again for the valuable feedback and we hope our response can address your questions.
> If you have any further questions or concerns, we are very happy to answer.
>
> [1] Wang, Xudong, et al. Cut and learn for unsupervised object detection and instance segmentation.
>
> [2] Uijlings, Jasper RR, et al. Selective search for object recognition.
>
> [3] Sowrirajan, Hari, et al. Moco pretraining improves representation and transferability of chest x-ray models.
>
> [4] He, Kaiming, et al. Masked autoencoders are scalable vision learners.
>
> [5] Xie, Zhenda, et al. Simmim: A simple framework for masked image modeling.

---

### Review · Reviewer_HCEj · 2025-07-20

**Summary Of Contributions:**

This manuscript analysis instance consistency based SSL methods (like MoCo-v2 and DINO) with regards to their depends on having true view-consistent positive pairs. This is particularly relevant in non-iconic datasets (in-the-wild datasets) where images contain multiple objects and two randomly cropped image regions (conventionally regarded as a positive pair) will likely not capture the same instance and thus not be a true positive pair.

The manuscript quantifies the extent to which a positive pair are similar using the earth mover distance. They use various cropping methods to get different data points on this spectrum (from highly consistent positive pairs with low earth mover distance to less consistent ones with high earth mover distance). Experiments show that too high view consistency and too low view consistency are bad for SSL pre-training. A sweet spot exists.

**Audience:**

Yes

**Broader Impact Concerns:**

This manuscript does not raise broader impact concerns beyond those of the SSL methods that it explores.

**Claims And Evidence:**

Yes

**Requested Changes:**

- Please expand the literature survey to include classic methods such as but not limited to the ones above.
- Discuss the metric being used in table 2
- Several of the strengths of the paper come from the sections in the supplementary. Why not use the full 12 pages?
- Table 2 middle block ("Baseline", "Spatial Ovlp. = 0", "Inst. vs Bg", "only bg") hasn't been referred to in the text. One option is to refer to these in section 4.1 "Results".
- Page 7 "Smaller Crop with Zero Spatial Overlap) consistently outperform the baseline" is probably a typo. The last row in in Table 1 has -4.13 for CIFAR 10 (ImageNet-100) for example.

**Strengths And Weaknesses:**

#### Strengths
- Experiments cover two pre-training objectives and several downstream tasks (classification, fine-grained classification, detection, instance segmentation, depth estimation).
- The presentation is easy to understand.

#### Weaknesses
- Literature review is sparse on methods before 2020. For example, jigsaw puzzles (https://arxiv.org/abs/1603.09246, ECCV 2016), RotNet (https://arxiv.org/abs/1803.07728, ICLR 2018), SSL methods using videos (eg. CVPR 2016 slow and steady feature analysis https://openaccess.thecvf.com/content_cvpr_2016/papers/Jayaraman_Slow_and_Steady_CVPR_2016_paper.pdf).
- An experiment that selects positive pairs based purely on the Earth mover distance is expected after the paragraph in page 9 "suggestions for positive pair selection". This experiment would help complete the experiments in this manuscript.

---

> ### Author Response · Authors · 2025-08-15
> **Response to Reviewer HCEj**
>
> We thank the reviewer for the constructive feedback and the insightful questions. We would like to address your concerns below:
>
> > [W1 C1] Expansion of Literature Survey to Classic SSL Methods.
>
> We appreciate the reviewer’s suggestion and acknowledge the value of covering foundational SSL methods.
>
> We will include the recommended citations and accompanying discussions in the **final version**.
>
> > [W2] Experiment on EMD-based Positive Pair Selection.
>
> We appreciate the reviewer’s suggestion and agree that a direct experiment selecting positive pairs based on EMD would strengthen our findings.
>
> In fact, our current findings already suggest a practical positive pair selection strategy in SSL: **pre-computing EMD scores before pre-training** and selecting positive pairs accordingly.
> Specifically, the optimal EMD range falls between the baseline configuration (used by default in MoCo-v2 and DINO) and the *Smaller Crop $^†$* configuration.
>
> This insight implies that one could first estimate an optimal EMD interval and use it to guide augmentation parameter selection or directly filter view pairs that satisfy the target EMD range prior to pre-training.
> Our experiment results with *Spatial Ovlp.=0* and *Smaller Crop* configurations also support the effectiveness of this strategy.
>
> We will include this discussion in Section 4.3 in the **final version**, and plan to explore direct EMD-based sampling in future work.
>
> > [C2 C4] Clarification on Experimental Setups.
>
> We appreciate the reviewer’s suggestion and will incorporate the requested clarifications in the **final version**.
>
> Specifically:
> - For detection tasks, we report the mean Average Precision (mAP) as the evaluation metric.
> - For the configurations listed in Table 1 and Table 2, we follow the definitions provided in Sections 4.1 and 4.2, which we will further elaborate for clear reference.
>
> > [C3] Extend the Paper to Full 12 Pages.
>
> We agree with the suggestion and will move the DINO results to the main paper in the **final version** to make better use of the available space.
>
> > [C5] Typo Correction.
>
> Thanks for pointing out this typo.
> We have updated the sentence in the **revised version** to:
>
> "configurations with higher diversity (i.e. *Smaller Crop*) consistently outperform the baseline."
>
> ---
> We thank the reviewer again for the valuable feedback and we hope our response can address your questions.
> If you have any further questions or concerns, we are very happy to answer.

---

### Review · Reviewer_xSmj · 2025-07-21

**Summary Of Contributions:**

This paper systematically revisits the instance‑consistency assumption in visual self‑supervised learning (SSL) that two augmented views of the same image must represent the same semantic instance. Through extensive experimentation, the authors empirically demonstrate that the instance‑consistency is not necessary to achieve strong downstream performance. The experiments cover two leading SSL frameworks (MoCo-v2 and DINO) and evaluate performance across diverse downstream vision tasks, including image classification, object detection, semantic segmentation, and depth estimation. The authors design controlled experiments that qualitatively vary the degree of instance-consistency by manipulating three factors: crop scale, spatial overlap between views, and the presence or absence of foreground objects.

To quantify the view diversity, the authors then propose Earth Mover’s Distance (EMD) as an estimator to measure mutual information between positive pairs. By systematically varying crop scales, the study explores how different degrees of view diversity impact downstream task performance. Their findings highlight an inverted-U relationship between EMD and linear probing accuracy on the COCO dataset. Empirical evidence thus suggests that moderate levels of view diversity (i.e., moderate EMD scores) yield optimal downstream results.

**Audience:**

Yes

**Broader Impact Concerns:**

No immediate ethical concerns are apparent.

**Claims And Evidence:**

Yes

**Requested Changes:**

I recommend the following major changes:
1. Weakness-1: Could you report confidence intervals or statistical tests for the key results?
2. Weakness-2: Could you discuss more on the potential bias introduced by using an external pre-trained encoder for EMD?

Additionally, I suggest some minor changes:
1. Weakness-4: The authors can evaluate sensitivity to mask quality by varying pseudo-mask generators or parameters.
2. In the current draft the backbone choice for the MoCo-v2 runs that feed Table 1 is only stated deep in Appendix A.1:
   > “For the backbone, we use ResNet-50 (He et al., 2016) in MoCo-v2 …”

   To avoid any ambiguity for readers who may not consult the appendix, I recommend adding this information in one of the following places: (1) Table 1 caption, or (2) somewhere in section 4.1.
3. The authors can explain Table 1 more clearly. Add two‑to‑three lines in section 4.2 interpreting the mixed results of Smaller Crop†.

**Strengths And Weaknesses:**

Strengths:
1. The authors conduct comprehensive experiments involving two SSL frameworks (MoCo-v2 and DINO), datasets (both iconic and non-iconic), and diverse downstream tasks (classification, detection, segmentation, depth estimation).
2. The authors introduce EMD as an estimator of view diversity and offer guidance for SSL training.
3. The empirical results provide clear implications for visual SSL involving non-iconic data.

Weakness:
1. The reported performance differences are relatively modest (Table 1 “Smaller Crop” & Table 2, typically 1-3% absolute), and no statistical significance tests or confidence intervals are provided. Prior augmentation-focused studies such as Object-aware cropping [1] achieve much larger margins (e.g. +8.8mAP on OpenImages), and SimCLR’s augmentation [2] ablations routinely yield more than 5 point swings. Without confidence intervals or significance tests, it is difficult to judge whether the observed gains are robust.
2. The authors did not explain the potential circularity in the EMD computation. EMD is computed on features from a pre‑trained encoder (ResNet-50). Earlier optimal‑transport SSL such as Self‑EMD performs distance computation within the training loop to avoid external feature bias [3]. Reliance on an external backbone can over‑estimate alignment quality, particularly if that backbone is architecturally similar to the model under evaluation. The observed empirical phenomena (e.g. U-shaped relationship) may not hold in general if we switch the encoder for EMD. For example, as reported in the DINO paper, a supervised ViT does not attend well to objects in presence of clutter both qualitatively and quantitatively.
3. The SSL methodology considered in the paper is limited. Adaptive mask-modeling approaches are straightforward solutions for non-iconic data, including MAE [4], SimMIM [5], AdaMAE [6]. They could eschew instance matching altogether while achieving competitive results. Excluding these paradigms weakens the generality of the conclusions.
4. The sensitivity and dependency to pseudo-mask quality is unaddressed. The ImageNet-100 experiments depend on the masks generated by MaskCut. For example, CutLER [7] shows that segmentation noise in such masks substantially affects downstream detection performance. An analysis of alternative mask generators or noise levels would clarify the robustness of the experiments.

References:

[1] Object-Aware Cropping for Self-Supervised Learning. 2021. Shlok Mishra, Anshul Shah, Ankan Bansal, Abhyuday Jagannatha, Janit Anjaria, Abhishek Sharma, David Jacobs, Dilip Krishnan.

[2] A Simple Framework for Contrastive Learning of Visual Representations. 2020. Ting Chen, Simon Kornblith, Mohammad Norouzi, Geoffrey Hinton.

[3] Self-EMD: Self-Supervised Object Detection without ImageNet. 2020. Songtao Liu, Zeming Li, Jian Sun.

[4] Masked Autoencoders Are Scalable Vision Learners. 2021. Kaiming He, Xinlei Chen, Saining Xie, Yanghao Li, Piotr Dollár, Ross Girshick.

[5] SimMIM: A Simple Framework for Masked Image Modeling. 2021. Zhenda Xie, Zheng Zhang, Yue Cao, Yutong Lin, Jianmin Bao, Zhuliang Yao, Qi Dai, Han Hu.

[6] AdaMAE: Adaptive Masking for Efficient Spatiotemporal Learning with Masked Autoencoders. 2022. Wele Gedara Chaminda Bandara, Naman Patel, Ali Gholami, Mehdi Nikkhah, Motilal Agrawal, Vishal M. Patel.

[7] Cut and Learn for Unsupervised Object Detection and Instance Segmentation. 2023. Xudong Wang, Rohit Girdhar, Stella X. Yu, Ishan Misra.

---

> ### Author Response · Authors · 2025-08-15
> **Response to Reviewer xSmj (1/2)**
>
> We thank the reviewer for the constructive feedback and the insightful questions. We would like to address your concerns below:
>
> > [W1 C1] Significance and Robustness of the Reported Performance Improvements.
>
> We appreciate the reviewer’s concern regarding both the significance and robustness of the reported gains.
>
> As discussed in **Appendix Section C.1**, prior works under similar settings typically report relatively modest gains when modifying views from an augmentation perspective.
> Our findings show that *targeted* control of view diversity can still yield improvements that are **practically significant**.
>
> For example, regarding the performance improvement in [1], when applying their approach under our VOC setting (as shown in [1]'s Table 6), the reported improvements are consistently around +1%, which is comparable to the gains we observe.
>
> Regarding the statistical significance analysis, all experiments in our study are repeated **three times**, and we report the average accuracy across these runs.
> Due to space constraints, we did not include the standard deviation values in the original submission.
>
> We have updated **standard deviations** for our main results (Figures 4 and B) in the **revised version**.
> These updated plots reflect the statistical variability and support the consistency of our findings across multiple runs.
>
> > [W2 C2] More Evaluations on EMD Computation.
>
> We appreciate the reviewer’s suggestion to further validate the robustness of EMD computation across different feature encoders.
>
> To address this, we provide a **cross-architecture validation** experiment where EMD is computed using various source feature encoders, and the results are correlated with the downstream performance of ResNet-based MoCo-v2 models.
>
> Specifically, we use the following **six feature encoders**: ResNet-50, ResNet-101, ViT-S, ViT-B, ViT-DINO-S, and ViT-DINO-B.
>
> As shown in **Appendix Figure C**, all six plots exhibit a clear and consistent reverse-U shape relationship, indicating that a moderate level of view diversity (as estimated via EMD) correlates with optimal downstream performance, regardless of the encoder used.
>
> This consistency across architectures further emphasizes the **generalizability** of our findings.
> We have included the above discussion in **Appendix Section B.5** in the **revised version**.
>
> > [W3] Applicability to Reconstruction-based SSLs.
>
> We appreciate the reviewer’s suggestion regarding the applicability of our findings to reconstruction-based SSLs.
>
> As clarified in **Section 2**, the focus of our work is to investigate *"the necessity of instance-level consistency in SSLs"*.
> This assumption is a central design principle in **contrastive- and distillation-based SSLs**, such as MoCo-v2 and DINO, which explicitly treat each image as a separate class.
> These methods primarily rely on consistency between augmented views to learn meaningful representations.
>
> In contrast, reconstruction-based SSLs like MAE [2] and SimMIM [3] are designed with a fundamentally different objective: to reconstruct the masked parts of an image, without requiring or benefiting from carefully designed view augmentations.
> As such, they do not rely on instance-level consistency and thus fall outside the scope of our study.
>
> For this reason, we focus our validation on contrastive- and distillation-based methods, which are the most representative paradigms aligned with our research objective.
>
> We hope the above clarification can address the reviewer's concern, and we have added the above discussion in **Appendix Section C.2** in the **revised version**.

---

> > ### Comment · Reviewer_xSmj · 2025-08-25
> >
> > Thanks for the response.
> >
> > [W1 C1] The added error bars and the note that each configuration is averaged over three runs are helpful.
> >
> > [W2 C2] The cross-architecture validation is helpful. To make the validation more actionable for practitioners, the authors can quantify the stability of the "sweet spot", including the peak location (EMD) and its 95% confidence interval.
> >
> > [W3] The scope clarification is reasonable.

---

> ### Author Response · Authors · 2025-08-15
> **Response to Reviewer xSmj (2/2)**
>
> > [W4 C3] Ablation Studies on Pseudo Mask Generators.
>
> We thank the reviewer for highlighting this important aspect.
>
> To further assess the robustness of our findings, we have conducted ablation experiments on pseudo mask generators for ImageNet, using two popular methods: MaskCut [4] and Selective Search [5].
> These results are presented in **Appendix Section B.4**, with further discussion regarding the implementation details included in **Appendix Section A.1**.
>
> Moreover, rather than relying on explicit bounding box or mask accuracy, we evaluate pseudo-mask quality via their impact on downstream SSL performance.
> Below, we show object detection results using MoCo-v2 pre-trained on COCO under three mask sources: ground truth (GT), MaskCut, and Selective Search.
>
> |VOC-0712|GT|MaskCut [4]|Selective Search [5]|
> |-|-|-|-|
> |Baseline|73.3|-|-|
> |Inst. vs Bg|74.2 (+0.9)|74.1 (+0.8)|74.2 (+0.9)|
> |Only Bg|74.7 (+1.4)|74.5 (+1.2)|74.6 (+1.3)|
>
> |DOTA-v1.0|GT|MaskCut [4]|Selective Search [5]|
> |-|-|-|-|
> |Baseline|54.4|-|-|
> |Inst. vs Bg|55.5 (+1.1)|55.7 (+1.3)|55.6 (+1.2)|
> |Only Bg|55.3 (+0.9)|55.3 (+0.9)|55.4 (+1.0)|
>
> As the results show, both pseudo-mask methods closely track the performance of ground-truth masks under diverse configurations and datasets.
> This supports our claim that the generated pseudo masks are sufficiently reliable to approximate instance locations for the purpose of our controlled analysis.
>
> We will add the above discussion and the corresponding experiments in the **final version**.
>
> > [C4 C5] Clarification on Experimental Setups and Reported Results.
>
> We thank the reviewer for pointing out the need for clearer clarification of our experimental setups and results.
>
> We will update the **final version** of the paper to incorporate the suggested modifications.
>
> ---
> We thank the reviewer again for the valuable feedback and we hope our response can address your questions.
> If you have any further questions or concerns, we are very happy to answer.
>
> [1] Mishra, Shlok, et al. Object-aware cropping for self-supervised learning.
>
> [2] He, Kaiming, et al. Masked autoencoders are scalable vision learners.
>
> [3] Xie, Zhenda, et al. Simmim: A simple framework for masked image modeling.
>
> [4] Wang, Xudong, et al. Cut and learn for unsupervised object detection and instance segmentation.
>
> [5] Uijlings, Jasper RR, et al. Selective search for object recognition.

---

> ### Comment · Reviewer_xSmj · 2025-08-25
> **Other questions**
>
> 1. Between pp. 1-6, the manuscript argues that strict instance consistency is not essential, but the term itself is not formally defined. In particular, zero spatial overlap between crops does not preclude both crops depicting the same physical instance (e.g., different, non-overlapping parts of a single large object). Could you elaborate more on the notion of instance consistency? This would help readers interpret the research background.
>
> 2. The authors can clarify what the line widths encode in figure 3. My understanding is that they visualize mass in the optimal-transport plan between patch features, but this is not stated explicitly. This would help readers interpret the figure.

---

> > ### Author Response · Authors · 2025-08-26
> >
> > Thank you for your feedback. We address your two follow-up points below:
> >
> > >[Q1] Clarifying the Notion of Instance Consistency.
> >
> > We appreciate the reviewer for highlighting the need to clarify the definition of *instance consistency*, which is central to our study.
> >
> > We adopt the term from the underlying assumption behind the commonly used instance discrimination pretext task in SSLs [1], where each training image is treated as a distinct semantic entity, and learning is driven by enforcing consistency between its augmented views.
> > In this context, *instance consistency* implies that consistent semantic information should be shared across views from the same image.
> >
> > Our work investigates the **necessity** of this assumption, especially when pre-training on non-iconic data, where such consistency may not naturally hold.
> >
> > We agree that *zero spatial overlap* between augmented views does not fully eliminate instance consistency, as non-overlapping regions of a large object instance may still share the same semantic information.
> > However, we would like to emphasize that this configuration only serves to **stepwise reduce** the degree of semantic consistency, achieving by firstly removing the spatial shared overlapping content between augmented views.
> > In practice, it offers a controlled setup to study the impact of different levels of instance consistency and its effect on SSL performance.
> >
> > We will highlight the above discussion in the description of the controlled ablation experiments in Experimental Setup in Section 4.1.
> >
> > >[Q2] Clarifying Line Widths in Figure 3.
> >
> > Thank you for pointing this out.
> > We have updated the following to the caption of Figure 3:
> >
> > "Different line widths between patches represent the mass transported under the OT plan used in EMD computation.
> > Thicker lines correspond to higher transport mass, indicating stronger correspondence between the two patches."
> >
> > ---
> > [1] Wu, Zhirong, et al. Unsupervised feature learning via non-parametric instance discrimination.

---

### Comment · Action_Editor_eR2y · 2025-06-19
**Correct in-text citations**

Dear authors,

I am happy to progress your paper to review, but before that, I would like to ask you to correct some of the in-text citations to make sure they follow our guidelines. For example: " More recently, (Morningstar et al., 2024) proposes a unified SSL framework" should use \citet".
Another example: "aligning with observations in (Van Gansbeke et al., 2021)" this should also use \citet.
"This aligns with findings from (Tian et al., 2020), "

There are a couple more, so please check all of them carefully and re-upload a revised version.

Please see the guidelines here:

"Citations within the text should be based on the natbib package and include the authors’ last names and year (with the “et al.” construct for more than two authors). When the authors or the publication are included in the sentence, the citation should not be in parenthesis, using \citet{} (as in “See Hinton et al. (2006) for more information.”). Otherwise, the citation should be in parenthesis using \citep{} (as in “Deep learning shows promise to make progress towards AI (Bengio & LeCun, 2007).”)."

Regards,

Action Editor

---

> ### Author Response · Authors · 2025-06-20
> **Revised version uploaded**
>
> Dear Action Editor,
>
> Thank you for your feedback. We have carefully reviewed all in-text citations throughout the paper and revised them to follow the TMLR citation guidelines. The revised version has been re-uploaded accordingly. Please let us know if any further adjustments are required.
>
> Best regards,
>
> The Authors

---

### Author Response · Authors · 2025-08-15
**General Comment on the Revised Version**

We sincerely thank the reviewers and the Action Editor for their valuable feedback.
As mentioned in our rebuttal, we have committed to incorporating several revisions, and we confirm the following:

1) **All updates promised in our rebuttal to be included in the "revised version" have been fully incorporated** into the updated submission.

2) **Other remaining points raised in the reviews have already been addressed in our rebuttal responses** and will be implemented in the *final camera-ready version*, primarily for better formatting.

Below is a summary of the updates made in the **current revised version**:

- **Added standard deviations** for main results, including Figure 4 and Appendix Figure B.

- **Added a cross-architecture validation experiment for EMD computation** in Appendix Section B.5, further strengthening the generalizability of our EMD-based estimator.

- **Corrected minor typos** throughout the manuscript.

- **Replaced all "fine-grained classification" with "classification"** to avoid potential confusion.

- **Added cross-references** from the main text to corresponding appendix sections to improve readability.

- **Added an ablation study on the impact of memory bank size in MoCo-v2** in Appendix Section B.6.

- **Added a discussion on the applicability of our findings to reconstruction-based SSLs** in Appendix Section C.2.

- **Added clarifications of the scope and contributions of our work** in Appendix Section C.3.

- **Reorganized the Appendix** by introducing Section C to group broader analytical insights by expanding original Section B.6.

---

> ### Comment · Action_Editor_eR2y · 2025-08-15
> **Thank you**
>
> Many thanks for your responses and the revision.
>
> I would like to invite the reviewers to take a look at all the responses and the revised manuscript and engage accordingly.
>
> Regards,
>
> AE

---

### Decision · Action_Editor_eR2y · 2025-09-03

**Recommendation:** Accept with minor revision

**Additional Comments:**

Please incorporate all changes promised in your responses to the reviewers.

**Audience:**

Yes

**Audience Explanation:**

This paper is highly relevant to TMLR and would also be useful to those working on fundamental SSL approaches and computer vision.

**Claims And Evidence:**

Yes

**Claims Explanation:**

This work presents a timely investigation into the foundational assumption of instance consistency in self-supervised learning (SSL). Traditionally, SSL frameworks rely on the premise that different augmented views of the same image represent the same semantic instance. This work challenges that assumption, especially in the context of non-iconic data, and demonstrates through extensive empirical analysis that meaningful representations can still be learned even when this consistency is relaxed.

The Earth Mover’s Distance (EMD) acts as a quantitative proxy for measuring view diversity, revealing a reverse-U relationship between EMD and downstream performance across classification, detection, segmentation, and depth estimation tasks. This insight is validated across multiple SSL paradigms (MoCo-v2, DINO), architectures (ResNet, ViT), and datasets. The paper also proposes a practical strategy for positive pair selection based on precomputed EMD scores, offering actionable guidance for SSL pipeline design.

The revised manuscript addresses earlier concerns with the addition of statistical analysis, cross-architecture validations, and broader experiments. The authors clarify the scope of their work, distinguishing it from prior studies such as InfoMin, and provide detailed responses to reviewers. While the reliance on a supervised encoder for EMD estimation introduces some limitations, the authors mitigate this with cross-encoder robustness checks and acknowledge the need for future work using label-free models.

Minor revisions:

Please incorporate all changes promised in the responses to reviewers, especially if these have not been added since submitting the revised document yet. An example "*We will highlight the above discussion in the description of the controlled ablation experiments in Experimental Setup in Section 4.1.*".